# RainPro-8: An Efficient Deep Learning Model to Estimate Rainfall Probabilities Over 8 Hours

**Rafael Pablos Sarabia**
Aarhus University & Cordulus
rpablos@cs.au.dk

**Joachim Nyborg**
Cordulus
jn@cordulus.com

**Morten Birk**
Cordulus
mb@cordulus.com

**Jeppe Liborius Sjørup**
Cordulus
jls@cordulus.com

**Anders Lillevang Vesterholt**
Cordulus
alv@cordulus.com

**Ira Assent**
Aarhus University
ira@cs.au.dk

## Abstract

We present a deep learning model for high-resolution probabilistic precipitation forecasting over an 8-hour horizon in Europe, overcoming the limitations of radar-only deep learning models with short forecast lead times. Our model efficiently integrates multiple data sources - including radar, satellite, and physics-based numerical weather prediction (NWP) - while capturing long-range interactions, resulting in accurate forecasts with robust uncertainty quantification through consistent probabilistic maps. Featuring a compact architecture, it enables more efficient training and faster inference than existing models. Extensive experiments demonstrate that our model surpasses current operational NWP systems, extrapolation-based methods, and deep-learning nowcasting models, setting a new standard for high-resolution precipitation forecasting in Europe, ensuring a balance between accuracy, interpretability, and computational efficiency. Code is available at https://github.com/rafapablos/RainPro.

## 1 Introduction

Recent advances in artificial intelligence have generated significant interest in deep learning for weather forecasting (Rasp et al., 2024; An et al., 2024). Deep learning excels at handling complex, large-scale, high-dimensional data, making it well-suited for capturing intricate, nonlinear patterns in spatio-temporal systems (Manzhu Yu & Li, 2024). Although deep learning has achieved remarkable success in both nowcasting (Gao et al., 2024b; Gong et al., 2024) and medium-range forecasting (Lam et al., 2023; Price et al., 2024), significant challenges remain. Deep learning models for nowcasting are often limited to very short lead times (up to two hours). In contrast, medium-range models, which predict broader atmospheric dynamics for up to 10 days, typically operate at coarser resolutions and are influenced by precipitation-specific biases in the training datasets (Lavers et al., 2022). As a result, they struggle to capture small-scale precipitation features, like local showers, often leading to the exclusion of precipitation forecasts in medium-range models (Lam et al., 2023).

This work addresses the challenge of forecasting precipitation for up to 8 hours at high spatio-temporal resolutions, bridging the gap between nowcasting and medium-range forecasting. Forecasting over an 8-hour horizon is critical for timely predictions that help mitigate risks like flooding and optimize resource management in agriculture, energy, or transportation. This task is particularly difficult due to the stochastic and sparse nature of precipitation, especially over extended lead times. Designing a deep learning model for this task requires addressing skewed and intermittent precipitation distributions, incorporating multi-sensor data, and ensuring robust uncertainty quantification.

Most existing deep learning-based nowcasting methods focus on generating radar-like precipitation forecasts (Ravuri et al., 2021; Gao et al., 2024b), but these become increasingly challenging and less accurate with longer lead times. Instead, probabilistic models emphasize uncertainty estimation and provide a more accurate and practical representation of rainfall at varying intensities. The MetNet

family of models (Sønderby et al., 2020; Espeholt et al., 2022; Andrychowicz et al., 2023) represents state-of-the-art deep learning-based systems for probabilistic precipitation forecasting. These models produce high-resolution forecasts for 8-24 hours in the United States, outperforming operational NWP systems. They achieve efficient probabilistic forecasting using cross-entropy loss on precipitation bins, requiring only one forward pass rather than an ensemble. However, this approach neglects the ordinality between bins. In addition, they rely on lead time conditioning, where the model generates forecasts for only one lead time at a time. While this approach is fully parallelizable across multiple GPUs, it significantly increases inference computational demands. Moreover, training MetNet requires substantial computational resources and is limited to the United States.

We propose RainPro-8, an efficient deep learning model to estimate **rain**fall **pro**babilities over **8** hours by determining the probability of different levels of precipitation at a given location and time. RainPro-8 is an efficient model based on MetNet-3 (Andrychowicz et al., 2023) that uses less than 20% of MetNet-3's training parameters to achieve 8-hour high-resolution precipitation forecasting across Europe — a region characterized by diverse climates, complex terrain, and highly variable precipitation dynamics (Ehmele et al., 2020). Its training objective explicitly accounts for the ordinality between precipitation bins. Lastly, it demonstrates that it is possible to generate all lead times simultaneously by downweighing later lead times during training, greatly enhancing inference efficiency and improving temporal consistency. RainPro-8 model shows substantial improvements over traditional precipitation forecasting methods, including NWP models and extrapolation-based techniques, as well as deep-learning nowcasting models, across various rain intensities for the next 8 hours. Evaluation on the widely used SEVIR (Veillette et al., 2020) benchmark also demonstrates strong and robust performance beyond 8-hour multi-source tasks. Our main contributions include:

- We propose an efficient neural architecture and training strategy designed to integrate multi-source data with varying temporal and spatial resolutions. The model produces consistent probabilistic predictions across all forecast lead times in a single forward pass, enabling efficient and improved forecasts.

- We conduct an extensive empirical evaluation demonstrating that RainPro-8 outperforms existing operational forecasting methods by 65% and achieves significant improvements over state-of-the-art deep learning nowcasting models. We further provide comprehensive ablation studies and model attribution analysis to quantify the impact of key design choices and input modalities.

- We demonstrate the versatility of RainPro by adapting it to radar-only 2-hour precipitation prediction on the SEVIR benchmark, where it achieves state-of-the-art performance compared to both deterministic and generative nowcasting approaches.

## 2 RELATED WORK

Traditional methods for weather forecasting are mostly numerical weather prediction (NWP) models, which simulate atmospheric dynamics using mathematical equations, also as ensembles over multiple simulations (Toth & Kalnay, 1997). NWP models demand significant compute, especially for ensembles, which restricts their spatial and temporal resolution. Their spin-up time results in poor performance for short lead times (Ma et al., 2021). Extrapolation-based methods, e.g. PySTEPS (Pulkkinen et al., 2019), RainyMotion (Ayzel et al., 2019) are limited to reduced forecast lengths due to their assumptions on constant motion and intensity (van Nooten et al., 2023).

Deep learning formulates precipitation nowcasting as a spatio-temporal prediction (An et al., 2024). Approaches in this area include the use of convolutions in recurrent networks (Shi et al., 2015; 2017b; Wang et al., 2023b; Ma et al., 2024), the U-Net architecture (Ayzel et al., 2020; Fernández & Mehrkanoon, 2021; Zhang et al., 2023; Trebing et al., 2021), transformers (Gao et al., 2024b; Wu et al., 2024), diffusion models (Leinonen et al., 2023; Gao et al., 2024a; Yu et al., 2024; Gong et al., 2024), and adversarial training (Ravuri et al., 2021). Some work improves loss functions to tackle data imbalance, blurriness, and sparsity in precipitation nowcasting models (Xu et al., 2024a; Ko et al., 2022; Yan et al., 2024). Still, most efforts focus on short-term forecasts (1 to 3 hours) using only radar, where reduced lead time eliminates the need for multiple data sources, large contexts, or accounting for higher uncertainty associated with 8-hour predictions.

Limited work exists for lead times beyond 3 hours. The Weather4Cast competition (Gruca et al., 2022) tackles 8-hour precipitation forecasting from satellite alone, but it has shown limited success compared to traditional methods (Li et al., 2023) due to challenges in estimating current precipitation from satellite imagery and resolution issues compared to radar. For 6-hour precipitation nowcasting, NPM (Park et al., 2024) relies solely on satellite data without data fusion, while Nowcast-to-Forecast (An, 2023) combines satellite and radar but is limited by a biased rainy-day dataset. Kim et al. (2024) only compare hourly forecasts to extrapolation baselines, which are ineffective for longer-term predictions. TAFFNet (Wang et al., 2023a) provides 12-hour predictions but neglects uncertainty and relies on radar and NWP data at equal resolutions not always available in practice.

MetNet-3 (Andrychowicz et al., 2023) is a leading deep learning model for precipitation forecasting, delivering 24-hour forecasts with high spatio-temporal resolution in the U.S. by integrating radar, weather station, satellite imagery, assimilated weather states, and other data in a transformer-based architecture. However, its training requires significant time and resources, involving hundreds of Tensor Processing Units (TPUs) for multiple days. Its reliance on high-quality data available only for the contiguous U.S. and the lack of public access to its code and data restrict its broader use. In this work, we show how to adapt its approach to problem formulation, data preprocessing, and training optimization for smaller and more efficient models and specific data requirements.

Medium-range forecasting models like GraphCast (Lam et al., 2023), Pangu-Weather (Bi et al., 2023), or GenCast (Price et al., 2024) rely on reanalysis datasets like ERA5 (Hersbach et al., 2020), which suffer from low resolution, delays, and biases, especially in surface variables and precipitation (Lavers et al., 2022). Precipitation, highly variable and challenging to predict, is underrepresented in such models (Rasp et al., 2024), with most failing to integrate critical observational data like radar. While some models, like NeuralGCM (Yuval et al., 2024), incorporate global precipitation predictions, they lack the high resolution and operational frequency needed for our setting. Other work (Xu et al., 2024b) has tried to address the gap between nowcasting and medium-range forecasting, but model performance drops at high temporal resolutions and focuses only on low precipitation.

## 3 RAINPRO-8 METHOD

### 3.1 PROBABILISTIC PRECIPITATION FORECASTING

Precipitation forecasting is a spatiotemporal prediction problem, where the goal is to predict $T_{out}$ future radar frames $Y$ based on a sequence of $T_{in}$ past radar frames $X$:

$$
\begin{aligned}
X &= [R_t]_{t=-T_{in}+1}^{0} \in \mathbb{R}^{T_{in} \times H \times W}, \\
Y &= [R_t]_{t=1}^{T_{out}} \in \mathbb{R}^{T_{out} \times H \times W},
\end{aligned}
\tag{1}
$$

where $R_t$ represents the precipitation intensity at timestep $t$ based on radar maps of size $H \times W$.

Radar data alone is insufficient for accurate precipitation forecasting over 8 hours because its ground-based systems limit its coverage and cannot capture atmospheric conditions beyond water vapor. Additional data sources like satellite or NWP offer broader coverage and a more comprehensive representation of the atmosphere. Such additional sources introduce challenges due to differences in temporal frequency and spatial resolution, requiring careful preprocessing and alignment as detailed in Appendix A. The generalized input $X$, with heterogenous sources *Sources*, is given by:

$$
X = \bigcup_{S \in Sources} [S_t]_{t=-T_{in,s}+1}^{0},
\tag{2}
$$

where $S_t \in \mathbb{R}^{C_s \times H_s \times W_s}$ represents a frame from the data source $S$, characterized by a specific number of channels, spatial dimensions, and resolution.

In addition to spatiotemporal accuracy, a critical priority are probabilistic forecasts: *What is the probability of a specific amount of rainfall at a given location and time?* Commonly, quantifying uncertainty involves generating multiple forecasts for ensembles to calculate probabilities, suffering from substantial computational demands. Instead of ensemble methods, our approach directly predicts the probability distribution of precipitation intensities, similar to MetNet 1-3 (Sønderby et al., 2020; Espeholt et al., 2022; Andrychowicz et al., 2023). Our goal is to generate accurate probability maps to capture uncertainty and variability in precipitation patterns, instead of radar-like outputs,

which become less reliable over longer lead times. To model probabilities for different precipitation intensity classes $I$, we redefine the target as probability maps for each precipitation intensity class:

$$Y = \bigcup_{t=1}^{T_{\text{out}}} \bigcup_{c=1}^{|I|} P_{t,c} \in \mathbb{R}^{T_{\text{out}} \times |I| \times H \times W}, \tag{3}$$

where $I$ is the set of intensity classes that divides the possible precipitation intensities into ranges or bins, and $P_{t,c}$ is the probability map for $R_t$ with respect to class $I_c$.

In MetNet, $P_{t,c} = P(R_t \in I_c)$ predicts the probability of precipitation intensity within predefined ranges, trained under cross entropy loss. However, this approach ignores the intrinsic order of the intensity classes, which we address with our proposed *Ordinal Consistent* loss.

## 3.2 ORDINAL CONSISTENT LOSS

To generate probabilistic forecasts that preserve ordinality among precipitation classes, we model $P_{t,c}$ as $P(R_t \geq \min(I_c))$. To ensure monotonicity ($P_{t,c} \leq P_{t,c-1}$), we reformulate $P_{t,c}$ using Bayes' theorem, given that $P(R_t \geq \min(I_{c-1})|R_t \geq \min(I_c)) = 1$, and redefine the model outputs:

$$\begin{aligned} P_{t,c} &= P(R_t \geq \min(I_c)) \\ &= \frac{P(R_t \geq \min(I_c)|R_t \geq \min(I_{c-1})) \times P(R_t \geq \min(I_{c-1}))}{P(R_t \geq \min(I_{c-1})|R_t \geq \min(I_c))} \\ &= P(R_t \geq \min(I_c)|R_t \geq \min(I_{c-1})) \times P_{t,c-1} \end{aligned} \tag{4}$$

Model output $P(R_t \geq \min(I_c)|R_t \geq \min(I_{c-1}))$ for each intensity class $c$ guarantees monotonic probabilities $P_{t,c}$ that respect the inherent order of precipitation classes. For the lowest intensity class, the model outputs $P(R_t \geq \min(I_1))$. This approach, as in Fernandes & Cardoso (2018) for semantic segmentation, improves both interpretability and consistency in the generated forecasts. For any class, probability $P_{t,c}$ is the cumulative product of model outputs for all preceding classes:

$$\begin{aligned} P_{t,c} &= P(R_t \geq \min(I_c) \mid R_t \geq \min(I_{c-1})) \times P_{t,c-1} \\ &= \prod_{j=2}^{c} P(R_t \geq \min(I_j) \mid R_t \geq \min(I_{j-1})) \times P(R_t \geq \min(I_1)) \end{aligned} \tag{5}$$

The loss uses target binary masks indicating whether $R_t \geq \min(I_c)$ for each class and timestep. The loss between the target masks and predicted probabilities is Binary Cross Entropy (BCE) without reduction to yield a per-pixel loss for each spatial location, class, and timestep. The ordinal consistent mask for every sample ($R_t \geq \min(I_{c-1})$) ensures that for each intensity class $c$, the loss is averaged only over pixels where the previous class is activated, encouraging the model to take advantage of class ordinality. Note that the loss is not averaged over prediction pixels that lack ground truth values ($R_t(h,w) = -1$), typically missing due to the limited coverage of radar data.

$$\mathcal{L} = \frac{1}{|S|} \sum_{(t,c,h,w) \in S} \text{BCE}(t,c,h,w), \quad \text{where } S = \{(t,c,h,w) : R_t(h,w) \geq \min(I_{c-1})\} \tag{6}$$

## 3.3 SINGLE-PASS PREDICTIONS

Precipitation nowcasting becomes significantly harder as lead times increase, which can negatively impact the model training. Auto-regressive models focus on one step at a time for stability but struggle when input sources differ from targets. MetNet tackles this with lead time conditioning, which supports multiple sources, but this approach significantly reduces inference efficiency and may cause temporal inconsistencies between lead times.

Instead, we make single-pass predictions to generate all forecast timesteps in one forward pass, reducing resource requirements, accelerating training convergence and inference, and improving temporal consistency in the outputs. The model encodes timestamps into the channel dimension, with the output $B(TC)HW$ reshaped to $BTCHW$, where $T$ represents all timesteps.

MetNet-3 uses lead time sampling for more frequent use of samples with shorter lead times during training. In contrast, we propose lead time weights, where all lead times are included in every training sample, but are weighted in the loss function to reduce the impact of longer lead times. This ensures that the model still prioritizes earlier lead times during training, and improves performance across the entire range of lead times, including the more challenging longer ones. Thus, we multiply the pixel-wise loss function by lead time weight $W_t$, normalized from an exponential distribution with decay rate $\alpha$, determining the relative weight of the first timestep over the last one.

$$\text{weights}_{\text{exp}}[t] = \exp(-\alpha \times t) \tag{7}$$

$$\text{weights}_{\text{norm}}[t] = \frac{\text{weights}_{\text{exp}}[t]}{\sum \text{weights}_{\text{exp}}[t]} \tag{8}$$

$$W_t = \frac{\text{weights}_{\text{norm}}[t]}{\overline{\text{weights}_{\text{norm}}}} \tag{9}$$

where $\text{weights}_{\text{exp}}$ is the weights based on exponential decay, $\text{weights}_{\text{norm}}$ is the normalized weights, and $W_t$ is the rescaled weights so the total loss scale remains consistent with the unweighted case.

## 3.4 ARCHITECTURE

We design our architecture to forecast precipitation for a $512 \times 512\text{km}^2$ patch. Assuming an average precipitation displacement rate of 1km per minute (Sønderby et al., 2020), we use 512km of spatial context on every side of the target patch ($1536^2\text{km}^2$) to provide our model with enough input context for all 8-hours of lead time. We use a U-Net (Ronneberger et al., 2015) with MaxViT (Tu et al., 2022) blocks to efficiently process multi-resolution data (Figure 1), similar to MetNet-3 (Andrychowicz et al., 2023). Key differences include single-pass predictions without lead time conditioning (Section 3.3), early downsampling in the encoder, halving internal channels, and removing topographical embeddings, all contributing to a reduced parameter count of 36.7M from the original 227M.

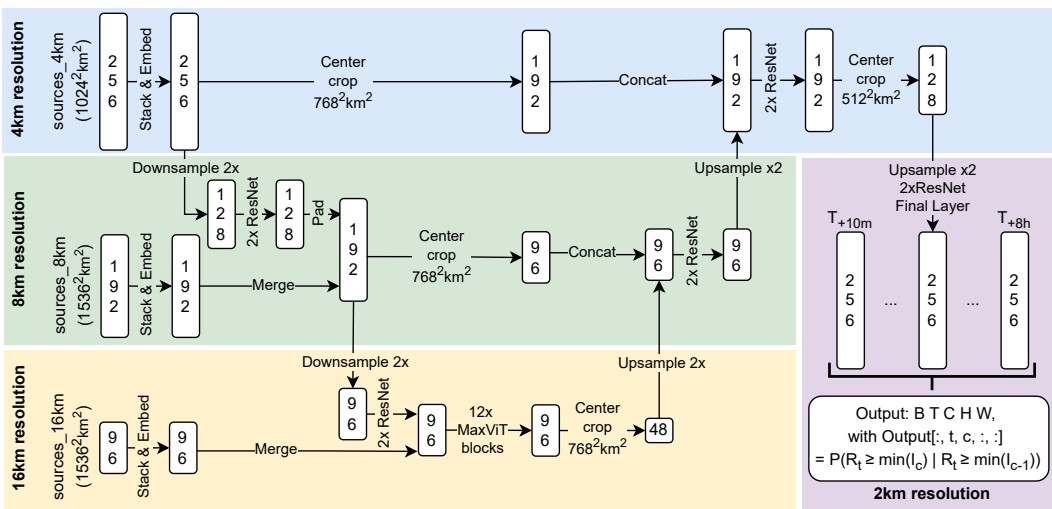

Figure 1: RainPro-8 architecture, optimized for reduced parameter count and efficiency, integrating multiple data sources of varying resolution for simultaneous prediction of all lead times.

The architecture begins with data fusion on the encoder, using Space-to-Depth convolutions (Sunkara & Luo, 2023) and ResNet blocks (He et al., 2016). Input data is transformed from $BTCHW$ to $B(TC)HW$ and merged with sources at matching resolutions. To balance efficiency and performance, we only use the full input context of 512km at 8km and 16km inputs, and lower input context of 256km at 4km resolution. The encoder aggregates features for each resolution into a low-resolution representation while preserving skip connections for the U-Net decoder. To improve memory efficiency, we apply downsampling before ResNet blocks instead of after.

The low-resolution representation is processed by 12 MaxViT blocks. Based on Vision Transformer (Dosovitskiy et al., 2021), they combine local neighborhood and global gridded attention aggregate information across the full $1536^2 \text{km}^2$ input patch. The decoder, using Transposed Convolutions (Long et al., 2015) and ResNet blocks, reconstructs high-resolution probability maps from the MaxViT output, while leveraging encoder skip connections. All forecast timesteps and intensity classes are generated by outputting $P(R_t \geq \min(I_c)|R_t \geq \min(I_{c-1}))$ (Section 3.2) in a single forward pass with timestamps encoded into the channel dimension (Section 3.3). For geographical alignment, representations are padded and cropped at each resolution. For example, 4km representations are padded equally on all sides before concatenating with 8km inputs, and MaxViT output is cropped to the target region with extra context for upsampling, later cropped to $512^2 \text{km}^2$ for the final output.

## 4   EXPERIMENTS

RainPro-8 is implemented with PyTorch (Ansel et al., 2024) and PyTorch Lightning (Falcon & team, 2025), trained for 100k steps using a batch size of 16, validating every 2,000 steps. Training uses a static learning rate of 3E-4, AdamW optimizer with a weight decay of 0.1 and betas $(0.9, 0.999)$, Exponential Moving Average (EMA) decay of 0.99975, dropout of 0.1, maximum stochastic depth (Huang et al., 2016) of 0.2, and lead time decay rate of 10. We use 256 channels throughout the entire network, totaling 36.7 million parameters. Any other network hyperparameters follow those of MetNet-3 (Andrychowicz et al., 2023). The model with the lowest validation loss is selected. Training is performed on an NVIDIA H100 SXM5 GPU and requires approximately 13 hours (cf. App. F).

We use multiple data sources as input to our precipitation forecasting model, each matched to its supported spatial resolutions. RainViewer[1] radar composites serve as high-resolution ground truth at 4km and 8km resolutions to capture local detail and broader context. Satellite imagery from EUMETSAT[2] provides cloud-related features at 8km resolution, and NOAA's GFS[3] provides atmospheric variables and precipitation forecasts at 16km resolution. Topographical information from Copernicus DEM[4] is incorporated at 4km resolution. To ensure consistency across these heterogeneous sources, all datasets are resampled to achieve spatial and temporal alignment. They further undergo normalization, clipping, and binning to handle varying scales and precipitation skewness. The data cover one year and over one million samples, with defined training, validation, and testing splits to ensure reliable model evaluation. Details on data, preprocessing, and samples in App. A.

We use Critical Success Index (CSI) (Schaefer, 1990) at different thresholds, a standard accuracy metric in precipitation nowcasting (Andrychowicz et al., 2023; Gao et al., 2024b; Gong et al., 2024), and Continuous Ranked Probability Score (CRPS) (Hersbach, 2000), which assesses alignment of predicted probability distribution with observed values, rewarding sharp and reliable predictions. Fractions Skill Score (FSS) (Roberts & Lean, 2008) accounts for intensity shifts and offers tolerance to translation and deformation. Frequency Bias Index (FBI) (Termonia et al., 2018) quantifies over- and underforecasting, but not forecast quality. We also report Mean Absolute Error (MAE) and Mean Squared Error (MSE), but note that they are sensitive to high frequency of no-rain cases and thus less suitable for skewed distributions (Andrychowicz et al., 2023) (cf. App. B.1). Taking inspiration from the thresholding approach in MetNet (Espeholt et al., 2022), intensity-based metrics use the mean of the highest activated bucket of the cumulative output distribution. A bucket is activated if its predicted probability mass exceeds the corresponding threshold, where thresholds are computed for each bucket and lead time using the validation set. This approach effectively captures rare high-precipitation events, which tend to have lower predicted probability masses.

---

[1] https://www.rainviewer.com/
[2] https://user.eumetsat.int/data/satellites/meteosat-second-generation
[3] https://registry.opendata.aws/noaa-gfs-bdp-pds
[4] https://registry.opendata.aws/copernicus-dem

Table 1: Performance (CSI, FSS), bias (FBI), error (MAE, MSE) across lead times and intensities.

| Model | Radar-only? | CSI ($\uparrow$) | FSS ($\uparrow$) | FBI ($\approx 1$) | MAE ($\downarrow$) | MSE ($\downarrow$) |
|---|---|---|---|---|---|---|
| RainPro-8 (ours) | $\times$ | **0.279** | **0.537** | 1.262 | 0.126 | 1.503 |
| MetNet-3* | $\times$ | 0.270 | 0.517 | 1.318 | 0.132 | 1.620 |
| GFS | $\times$ | 0.110 | 0.253 | 0.780 | 0.164 | 1.453 |
| PySTEPS | $\checkmark$ | 0.149 | 0.364 | **0.983** | 0.162 | 2.324 |
| RainPro-8R | $\checkmark$ | 0.229 | 0.449 | 1.346 | 0.144 | 1.735 |
| Earthformer | $\checkmark$ | 0.111 | 0.267 | 0.163 | **0.110** | 1.358 |
| SimVP | $\checkmark$ | 0.122 | 0.287 | 0.189 | 0.118 | **1.340** |

## 4.1 PERFORMANCE EVALUATION

RainPro-8 is benchmarked against global NWP GFS[5], and PySTEPS (Pulkkinen et al., 2019) which extrapolates radar echoes with a 512 km context. GFS forecasts are bilinearly interpolated to 2 km/px and temporally aligned to 10 min for radar comparison. No other deep-learning nowcasting model can learn from all our input sources: Earthformer (Gao et al., 2024b) and SimVP (Gao et al., 2022) operate only on radar with fixed resolution and coverage, and other data sources cannot be handled by their architectures. For fairness, all models use the same input region (with added context) but are evaluated only on the central target region, including a radar-only RainPro-8 variant (RainPro-8R) to demonstrate that our approach provides further performance benefits beyond its multiple data source capabilities. SimVP, which produces outputs matching its input length, is run autoregressively with inputs interpolated from 4 km to 2 km to achieve the target resolution. Earthformer is trained at 4 km due to attention limits, with outputs upsampled to 2 km for evaluation. MetNet-3 (Andrychowicz et al., 2023) is challenging to evaluate due to its private code, reliance on US-specific data, and substantial computational requirements (227M parameters trained on 512 TPU v3 cores over 7 days). MetNet-3* is our faithful reimplementation of the architecture and training described in the original paper, adapted to our data and compute constraints, presenting a reproducible competitor under fair conditions. It incorporates lead time conditioning and cross-entropy loss, and it only computes the loss on the high-resolution precipitation maps, not on accumulated rain or NWP initial state since we do not perform densification from weather stations.

As shown in Table 1, RainPro-8 outperforms all competitors on both precipitation metrics, CSI and FSS. RainPro-8 performs slightly better than MetNet-3*, and offers additional advantages: a $48\times$ inference speedup through single-step prediction and more coherent probability maps, enabled by the ordinal-consistent loss that accounts for class ordinality in probabilistic forecasts. RainPro-8R surpasses existing radar-only baselines, demonstrating that our approach offers superior performance, but also lagging behind RainPro-8 at longer lead times, underscoring the benefit of our multi-source integration in our full model. FBI indicates that RainPro-8 slightly overpredicts precipitation, Earthformer and SimVP significantly underpredict, and PySTEPS is the least biased. Although SimVP and Earthformer achieve lowest MAE and MSE due to their MSE loss, these do not translate into higher CSI or FSS, likely suffering from impact of the dominating no-rain class.

Figure 2 illustrates model performance (CSI) at different rain intensities and lead times up to eight hours (further details in App. B.2). Our model demonstrates superior performance across lead times and rainfall thresholds, from light to heavy rain. The skill gap with extrapolation-based PySTEPS grows as lead times increase due to its assumption of constant motion and intensity. The skill gap also widens with lead time, where Earthformer and SimVP lack sufficient input and probabilistic output capabilities. RainPro-8R confirms the impact of lack of additional data sources over time. In contrast, GFS shows a skill gap at shorter lead times because of its longer convergence time. Performance is slightly lower for all models at higher precipitation intensities, likely due to complexity in forecasting extreme weather, which often involves more chaotic and unpredictable patterns.

Figure 3 visualizes a $1000^2 km^2$ cropped region of at least 4 target patches of $512^2 km^2$, at 3 lead times (Europe-wide in App. D). RainPro-8 effectively captures areas with rainfall, both in intensity and location, with some blurriness as lead time increases uncertainty. GFS shows major deviations

---

[5]https://registry.opendata.aws/noaa-gfs-bdp-pds

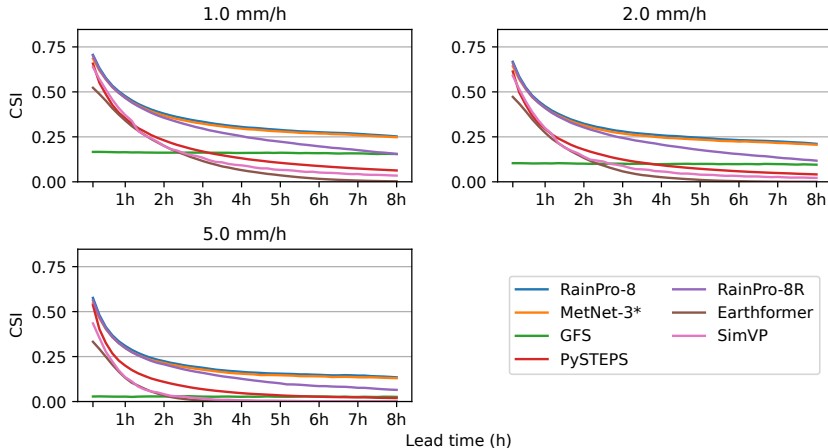

Figure 2: Critical Success Index (CSI) across different thresholds and lead times.

from ground truth, with large differences in intensity and rain-covered areas. PySTEPS struggles to capture changes in intensity or account for non-linear motion patterns at later lead times as it only extrapolates the latest radar image. RainPro-8R, Earthformer, and SimVP cannot exploit additional data, and the latter two suffer from MSE loss favoring no-rain events.

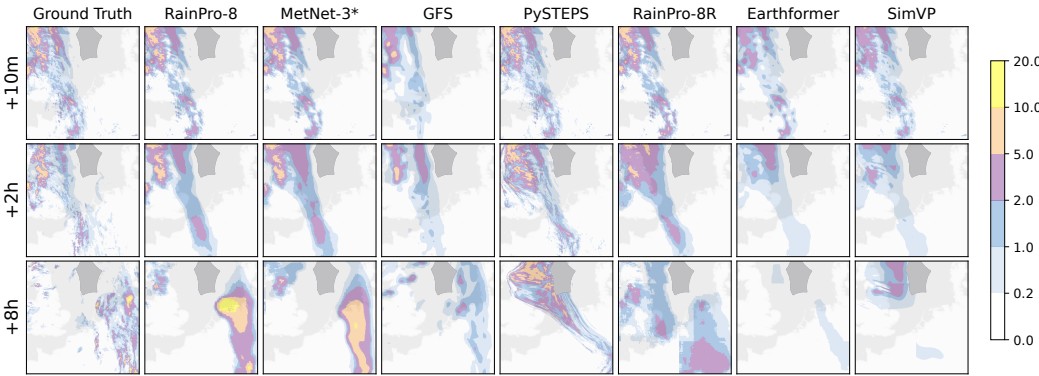

Figure 3: Sample ground truth and forecasts for different models at selected lead times with origin at 2024-01-23 11:20 UTC in cropped region. Dark grey areas indicate regions beyond radar coverage.

## 4.2 ABLATION STUDIES

Table 2 shows ablation studies of our ordinal-consistent loss, single-pass predictions, lead time weighting and multi-source input, compared to MetNet-3* using non-ordinal cross-entropy loss and lead time conditioning (Andrychowicz et al., 2023). Results confirm the effectiveness of our approach, and the contribution of each of our design choices and learning strategies. The slight yet consistent gains of our model over the cross-entropy-based model (cf. App G) are due to the latter enforcing ordinality only at inference time via cumulative probability computation, rather than learning it during training. Compared to models using lead time conditioning, our single-pass prediction not only improves inference speed (cf. App F) but also delivers superior performance across metrics. Excluding lead time weights leads to a noticeable drop in performance. Finally, an architecture capable of integrating all available data is key to top accuracy compared to radar-only RainPro-8R.

Table 2: Ablation loss function, timesteps per forward pass, lead time weights, data sources.

| Model | $\mathcal{L}$ | $T_{size}$ | $OC_{ltw}$ | Srcs. | CRPS ($\downarrow$) | CSI ($\uparrow$) | FSS ($\uparrow$) |
|---|---|---|---|---|---|---|---|
| RainPro-8 (ours) | OC | 48 | $\checkmark$ | $\checkmark$ | **0.06096** | **0.2791** | **0.5367** |
| CE loss (no OC) | CE | 48 | $\checkmark$ | $\checkmark$ | 0.06098 | 0.2787 | 0.5357 |
| Lead time conditioning | OC | 1 | - | $\checkmark$ | 0.06203 | 0.2695 | 0.5191 |
| No lead time weights | OC | 48 | $\times$ | $\checkmark$ | 0.06156 | 0.2733 | 0.5258 |
| Radar-only (RainPro-8R) | OC | 48 | $\checkmark$ | $\times$ | 0.06574 | 0.2289 | 0.4491 |
| MetNet-3* | CE | 1 | - | $\checkmark$ | 0.06199 | 0.2697 | 0.5173 |

## 4.3 PROBABILITY MAPS

RainPro-8 not only outperforms baselines in precipitation intensity forecasts but also quantifies uncertainty for each intensity. Figure 4 displays probability maps and ground truth for two rain intensities and lead times (same sample and crop as Figure 3), with intensities derived from these probabilities. Uncertainty is higher for longer lead times and light rain, reflecting the challenge of predicting scattered light rain far into the future at high resolution. Higher intensities are predicted with lower probabilities because of their low likelihood, particularly at extended lead times. The model ensures consistency by never assigning higher probabilities to higher precipitation levels.

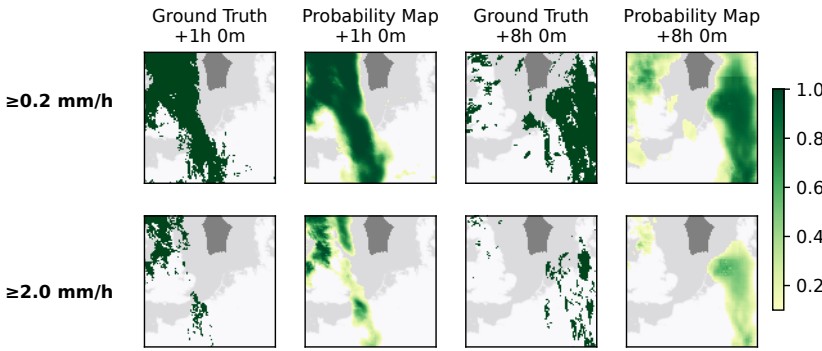

Figure 4: Ground truth and RainPro-8 probability map for different rain intensities and lead times, origin at 2024-01-23 11:20 UTC. Dark grey areas indicate regions beyond radar coverage.

## 4.4 ATTRIBUTION

We analyze RainPro-8 with Integrated Gradients (IG) (Sundararajan et al., 2017), as in MetNet-2 (Espeholt et al., 2022), attributing predictions to inputs to reveal how data sources are used. Rather than isolating each source with separate experiments, we use IG to assess their relative importance across lead times and variables (cf. App. C). Results show recent high-res radar drives short lead times, while low-res radar gains value around 4 hours for broader coverage. Beyond 4 hours, satellite data becomes more useful, though visible channels contribute little due to daytime limits. GFS variables become more influential for longer lead times, with key contributors including wind components, vertical velocity, storm motion parameters, specific humidity, and surface metrics such as pressure and the Best Lifted Index (a measure of atmospheric instability), which are particularly relevant for forecasting cloud movement, precipitation, and thunderstorms. GFS forecasts become increasingly impactful at longer lead times, while less relevant for the first 4 hours.

## 4.5 RADAR-ONLY SHORT-TERM NOWCASTING (SEVIR BENCHMARK)

While RainPro-8 targets 8-hour high-resolution precipitation forecasts using multi-source data, we also evaluate a radar-only short-term version (RainPro-2R) for 2-hour predictions on the SEVIR benchmark for direct comparison with existing nowcasting models (App. H). Table 3 shows

RainPro-2R outperforms state-of-the-art deterministic models. Compared to generative models, it achieves higher CSI and HSS (pixel-wise metrics) but lower pooled CSI and perceptual metrics (LPIPS, SSIM), reported in Yu et al. (2024) (App. H). Pooled CSI only checks whether any pixel within a neighborhood is hit, ignoring how many pixels match, whereas FSS rewards both correct pixels and the fraction of pixels correctly predicted, giving a more nuanced measure of spatial accuracy at full resolution (Figure 5). Perceptual metrics reflect a trade-off: sharper outputs sacrifice per-pixel performance (Yan et al., 2024). RainPro-2R's probabilistic predictions appear blurrier but capture uncertainty and higher accuracy. RainPro-2R also outperforms DiffCast on CRPS and FSS while being 13× faster at inference. These results highlight RainPro's performance both for the targeted 8-hour multi-source forecasting and SEVIR's short-term radar-only nowcasting setting.

Table 3: SEVIR benchmark; best results in bold per category; * from DiffCast (Yu et al., 2024)

| Method | | CSI (↑) | CSI-p4 (↑) | CSI-p16 (↑) | HSS (↑) | LPIPS (↓) | SSIM (↑) |
|---|---|---|---|---|---|---|---|
| Deterministic | RainPro-2R | **0.3524** | **0.3834** | **0.4171** | **0.4501** | **0.2353** | 0.5966 |
| | PhyDnet* | 0.2560 | 0.2685 | 0.3005 | 0.3124 | 0.3785 | 0.6764 |
| | MAU* | 0.2463 | 0.2566 | 0.2861 | 0.3004 | 0.3933 | 0.6361 |
| | ConvGRU* | 0.2416 | 0.2554 | 0.3050 | 0.2834 | 0.3766 | 0.6532 |
| | SimVP* | 0.2662 | 0.2844 | 0.3452 | 0.3369 | 0.3914 | 0.6304 |
| | Earthformer* | 0.2513 | 0.2617 | 0.2910 | 0.3073 | 0.4140 | **0.6773** |
| Generative | STRPM* | 0.2512 | 0.3243 | 0.4959 | 0.3277 | 0.2577 | **0.6513** |
| | MCVD* | 0.2148 | 0.3020 | 0.4706 | 0.2743 | 0.2170 | 0.5265 |
| | PreDiff* | 0.2304 | 0.3041 | 0.4028 | 0.2986 | 0.2851 | 0.5185 |
| | DiffCast* | **0.3077** | **0.4122** | **0.5683** | **0.4033** | **0.1812** | 0.6354 |

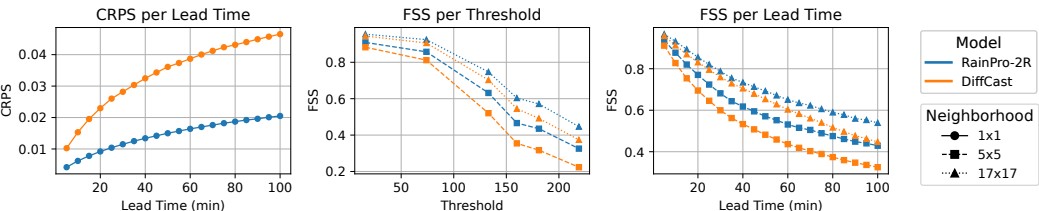

Figure 5: CRPS and FSS across different neighborhoods, thresholds, and lead times.

## 5 CONCLUSION & FUTURE WORK

We introduce RainPro-8, a deep learning model for precipitation nowcasting that outperforms existing operational systems and deep learning nowcasting models. RainPro-8 addresses the challenges of forecasting probabilistic predictions at different precipitation levels. It leverages multiple data sources with an efficient model that forecasts all lead times simultaneously, exploiting ordinality of precipitation levels. Going forward, RainPro-8 could be extended to explicitly address robustness to missing data which is of particular importance in operational settings where some input source could become absent, and to automatically learn thresholds.

ACKNOWLEDGMENTS

This work is partly funded by the Innovation Fund Denmark (IFD) under File No. 2052-00064B. Computational resources were provided by Lambda and the GenomeDK cluster at Aarhus University.

REPRODUCIBILITY STATEMENT

The code is publicly available at `https://github.com/rafapablos/RainPro`, along with instructions for accessing data via the official sources. The paper provides comprehensive details on the RainPro-8 framework, architecture, and evaluation, while the appendices serve as supporting material. Specifically, Section 3 describes the task definition, loss function, output generation, and model architecture. Section 4 summarizes the training setup and hyperparameters (with further details in Appendices E and F), data sources (including preprocessing and sample details in Appendices A and I), and baselines, metric definitions, and performance comparisons (expanded in Appendices B and G). Section 4 also presents visualizations (also in Appendix D), ablation studies, attribution analyses (also in Appendix C), and an evaluation of a RainPro-8 variant adapted for the benchmark task on a standard precipitation nowcasting benchmark (also in Appendix H).

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

# A  DATASET

## A.1  DATA SOURCES

We use RainViewer[6] radar composite data over Europe, which consists of high-resolution rainfall intensity measurements at 10-minute intervals and a resolution of 2 kilometers per pixel. Geostationary satellite data is obtained from the European Organisation for the Exploitation of Meteorological Satellites (EUMETSAT)[7], capturing cloud-related characteristics that significantly influence precipitation, despite its lower resolution (3-11km) and the challenges of correlating satellite and radar data (Stock et al., 2024). Topographical data is retrieved from the Copernicus Digital Elevation Model (DEM)[8]. Atmospheric observations and derived physical parameters are sourced from the Global Forecast System (GFS)[9], managed by the National Oceanic and Atmospheric Administration (NOAA) in the United States, with a spatial resolution of 28km and update frequency of 6 hours. In contrast, MetNet (Andrychowicz et al., 2023) uses HRRR, with 3km resolution, hourly updates, and US-only coverage.

## A.2  DATA PREPROCESSING

Each data source is modified to match the desired resolution for model input, either through upsampling, such as GFS data from 28km/px to 16km/px, or downsampling, like radar data from 2km/px to both 4km/px and 8km/px. The time dimension also presents challenges to consider. Satellite data, for instance, has a 1-hour operational delay, making the most recent timestep available at -60 minutes. For the source *gfs_16km*, we select 122 channels representing multiple weather variables at different pressure levels (Appendix I). As GFS is only initialized 4 times a day, we use the latest forecast lead time as our initial state. The subsequent lead times form the *gfs_forecast_16km*, from which we use only the precipitation rate variable. Table 4 summarizes all data sources with the corresponding spatial and temporal details.

Table 4: Outputs and inputs for the precipitation forecasting model.

| Variable | Source | Size (px) | Res. (km/px) | Size (km) | Timesteps (min) | Channels |
|---|---|---|---|---|---|---|
| target_2km | RainViewer | 256 | 2 | 512 | [10,20,...,480] | 1 |
| radar_4km | RainViewer | 256 | 4 | 1024 | [-60,-50,...,0] | 1 |
| radar_8km | RainViewer | 192 | 8 | 1536 | [0] | 1 |
| satellite_8km | EUMETSAT | 192 | 8 | 1536 | [-120,-105,...,-60] | 11 |
| gfs_16km | GFS | 96 | 16 | 1536 | [0] | 122 |
| gfs_forecast_16km | GFS | 96 | 16 | 1536 | [60,120,...,480] | 1 |
| xyz_4km | DEM | 256 | 4 | 1024 | N/A | 3 |
| minute_4km | - | 256 | 4 | 1024 | N/A | 1 |

We tackle challenges such as varying data ranges and skewed distribution of precipitation intensities using normalization, clipping, and binning. Min-max normalization is applied to all data sources, as many variables are non-normally distributed. Radar reflectivity values (dBZ) are clipped to a range of -1 to 64 to remove outliers. To address rainfall skewness, target radar maps are categorized into predefined rain intensity classes for multi-label classification. Using the Marshall-Palmer equation (Marshall & Palmer, 1948), dBZ values are converted to mm/h, with finer granularity for light rain ($\geq 0.1, \geq 0.2, \geq 0.4$) and broader ranges for heavier rainfall (up to $\geq 25.0$ mm/h) (Appendix E). This approach captures subtle light rain variations while accounting for rare intense precipitation events, aligning with the original dBZ distribution.

---

[6]https://www.rainviewer.com/
[7]https://user.eumetsat.int/data/satellites/meteosat-second-generation
[8]https://registry.opendata.aws/copernicus-dem
[9]https://registry.opendata.aws/noaa-gfs-bdp-pds

### A.3 Split Generation

Data spans from December 2023 to November 2024 and is partitioned into training, validation, and testing sets using multi-day cycles (12, 2, and 2 days, respectively). A 12-hour blackout period between cycles prevents data leakage.

Samples are generated using a sliding window along the temporal dimension for non-overlapping $512 \times 512 \text{km}^2$ patches, ensuring alignment with the timesteps and spatial dimensions in Table 4. The training set includes 1,063,658 samples, each with at least 50% radar coverage. During training, we apply uniform random offsets of ±256km horizontally and vertically to patches to make the training dataset as large as possible. Validation and test sets comprise 5,120 and 15,360 randomly selected samples, respectively, including challenging cases with radar coverage below 50%. Although evaluation is limited to pixels with radar coverage, boundary regions remain more challenging to predict due to the absence of nearby radar coverage.

# B METRICS

## B.1 METRIC DEFINITIONS

**Critical Success Index (CSI)** CSI, also known as the Threat Score, evaluates the accuracy of event detection by comparing the correctly predicted precipitation events to all events that were either predicted or actually occurred. It balances misses and false alarms, making it particularly useful in rare-event forecasting like heavy rainfall. A value of 1 indicates perfect accuracy, while 0 indicates no skill.

$$\text{CSI} = \frac{\text{TP}}{\text{TP} + \text{FP} + \text{FN}}, \tag{10}$$

where TP, FP, and FN are true positives, false positives, and false negatives, based on the threshold (rate $\geq$ threshold). The thresholds at which the metric is evaluated are 0.5, 1, 2, 5, and 10 mm/h.

**Continuous Ranked Probability Score (CRPS)** CRPS measures the accuracy of the entire predicted cumulative distribution function (CDF) relative to the observed outcome. It generalizes MAE for probabilistic forecasts, with lower values indicating better forecast reliability, sharpness, and calibration. It rewards predictions that assign high probability to the correct intensity range.

$$\text{CRPS} = \sum_{c=1}^{|I|} [(P(R_t < \min(I_c)) - \mathbb{1}(R_t < \min(I_c))]^2 \times |I_c|, \tag{11}$$

where $c$ iterates over all intensity classes, $\min(I_c)$ is the lower end of class $c$, $R_t$ is the target radar, and $P(R_t < \min(I_c)) = 1 - P(R_t \geq \min(I_c))$, based on the model probability $P_{t,c}$.

**Fraction Skill Score (FSS)** FSS provides an estimate of spatial accuracy by comparing the forecast and observed precipitation fields over a local neighborhood, rather than point-wise accuracy. FSS is particularly useful for evaluating high-resolution forecasts, as it accounts for small spatial displacements in predicted precipitation. A perfect forecast yields an FSS of 1, while lower values indicate poorer spatial agreement.

$$\text{FSS} = 1 - \frac{\sum_{i=1}^{H} \sum_{j=1}^{W} (F_{i,j} - O_{i,j})^2}{\sum_{i=1}^{H} \sum_{j=1}^{W} F_{i,j}^2 + \sum_{i=1}^{H} \sum_{j=1}^{W} O_{i,j}^2}, \tag{12}$$

where $F_{i,j}$ and $O_{i,j}$ refer to the fraction of predicted positives and fraction of observed positives, respectively, in the neighborhood of the $(i, j)$ pixel. This metric is computed at different thresholds (0.5, 1, 2, 5, and 10 mm/h) and neighborhood sizes (2km, 10km, and 20km). This metric also requires binarization based on the computed thresholds.

**Frequency Bias Index (FBI)** FBI quantifies whether a model tends to overforecast or underforecast precipitation events. A value greater than 1 indicates overprediction, while a value less than 1 indicates underprediction. While a bias of 1 is ideal, it doesn't imply the forecast is accurate—just that the frequency of forecasted events matches the observed frequency.

$$\text{FBI} = \frac{\text{TP} + \text{FP}}{\text{TP} + \text{FN}}, \tag{13}$$

where TP, FP, and FN are true positives, false positives, and false negatives, based on the threshold (rate $\geq$ threshold). The thresholds at which the metric is evaluated are 0.5, 1, 2, 5, and 10 mm/h.

## B.2 Additional Evaluation

Table 5: Critical Success Index (CSI) across different thresholds for precipitation forecasting models. Our model RainPro-8 outperforms all other models across different rain intensities.

| | CSI (↑) | | | | |
|---|---|---|---|---|---|
| Model | 0.2 | 1.0 | 2.0 | 5.0 | 10.0 |
| RainPro-8 (ours) | **0.447** | **0.344** | **0.296** | **0.204** | **0.146** |
| MetNet-3* | 0.437 | 0.335 | 0.287 | 0.194 | 0.136 |
| GFS | 0.253 | 0.161 | 0.100 | 0.027 | 0.009 |
| PySTEPS | 0.271 | 0.180 | 0.141 | 0.090 | 0.062 |
| RainPro-8R | 0.398 | 0.292 | 0.245 | 0.164 | 0.112 |
| Earthformer | 0.265 | 0.125 | 0.088 | 0.039 | 0.015 |
| SimVP | 0.256 | 0.154 | 0.116 | 0.047 | 0.023 |

Table 6: Fraction Skill Score (FSS) across different thresholds for precipitation forecasting models with a neighborhood size of 10km. Our model RainPro-8 outperforms all other models across different rain intensities.

| | FSS-10km (↑) | | | | |
|---|---|---|---|---|---|
| Model | 0.2 | 1.0 | 2.0 | 5.0 | 10.0 |
| RainPro-8 (ours) | **0.728** | **0.639** | **0.590** | **0.469** | **0.363** |
| MetNet-3* | 0.716 | 0.625 | 0.572 | 0.445 | 0.336 |
| GFS | 0.489 | 0.370 | 0.268 | 0.101 | 0.041 |
| PySTEPS | 0.552 | 0.430 | 0.369 | 0.283 | 0.225 |
| RainPro-8R | 0.665 | 0.554 | 0.496 | 0.378 | 0.275 |
| Earthformer | 0.509 | 0.279 | 0.211 | 0.116 | 0.056 |
| SimVP | 0.485 | 0.338 | 0.277 | 0.134 | 0.075 |

Table 7: Frequency Bias Index (FBI) across different thresholds for precipitation forecasting models. Our model, RainPro-8, has a tendency to over-predict high-precipitation events, in contrast to GFS, Earthformer, and SimVP, which consistently under-predict rainfall. PySTEPS appears the most balanced in terms of bias, with over- and under-predictions occurring at similar rates—though this does not translate to higher forecast accuracy.

| | FBI (≈ 1) | | | | |
|---|---|---|---|---|---|
| Model | 0.2 | 1.0 | 2.0 | 5.0 | 10.0 |
| RainPro-8 (ours) | 1.140 | 1.137 | 1.142 | 1.268 | 1.636 |
| MetNet-3* | 1.124 | 1.136 | 1.160 | 1.364 | 1.821 |
| GFS | 1.602 | 1.177 | 0.700 | 0.292 | 0.127 |
| PySTEPS | **0.899** | **0.995** | **0.914** | **1.029** | **1.078** |
| RainPro-8R | 1.131 | 1.141 | 1.180 | 1.356 | 1.932 |
| Earthformer | 0.397 | 0.182 | 0.135 | 0.063 | 0.027 |
| SimVP | 0.384 | 0.244 | 0.197 | 0.074 | 0.034 |

The following plots expand on the previous tables in terms of lead times.

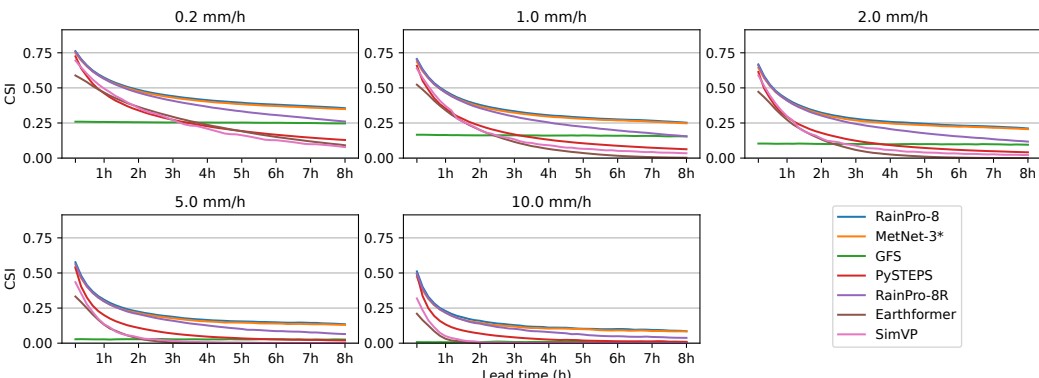

Figure 6: Critical Success Index (CSI) for Europe-wide precipitation forecasting models across different thresholds and lead times.

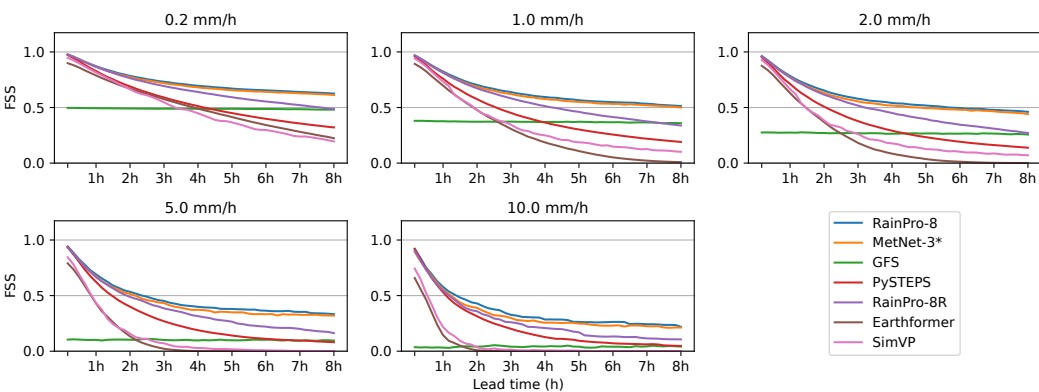

Figure 7: Fraction Skill Score (FSS) for Europe-wide precipitation forecasting models across different thresholds and lead times with neighborhood size of 10km.

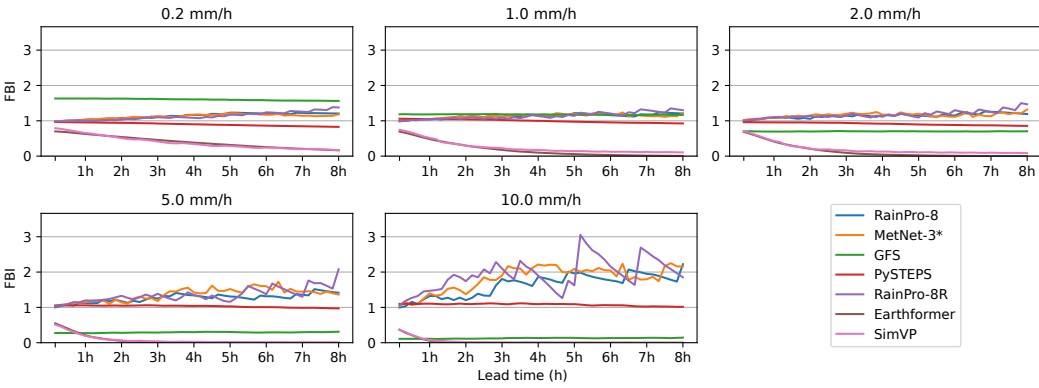

Figure 8: Frequency Bias Index (FBI) for Europe-wide precipitation forecasting models across different thresholds and lead times.

## C  INPUT ATTRIBUTION WITH INTEGRATED GRADIENTS

The following plots show the results obtained on the input attribution by using Integrated Gradients from which the key findings were obtained in Section 4.4. Integrated Gradients estimates feature importance by interpolating between a baseline and the actual input, averaging model gradients along this path. The final attribution is obtained by scaling these averaged gradients by the input difference, quantifying each feature's contribution to predictions. To ensure meaningful attribution, we use the minimum value of each feature as the baseline, as this results in near-zero precipitation probabilities. We then aggregate feature attributions across space, time, and samples, yielding a single importance score per feature.

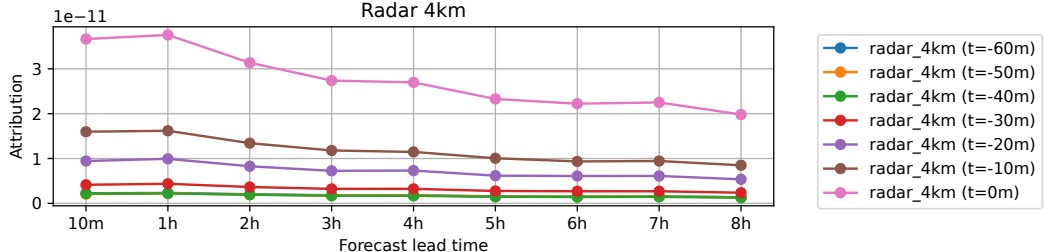

Figure 9: Attribution of radar_4km input timesteps over forecast lead time.

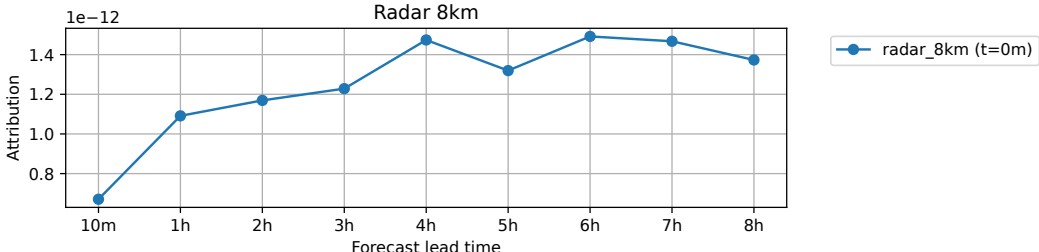

Figure 10: Attribution of radar_8km over forecast lead time.

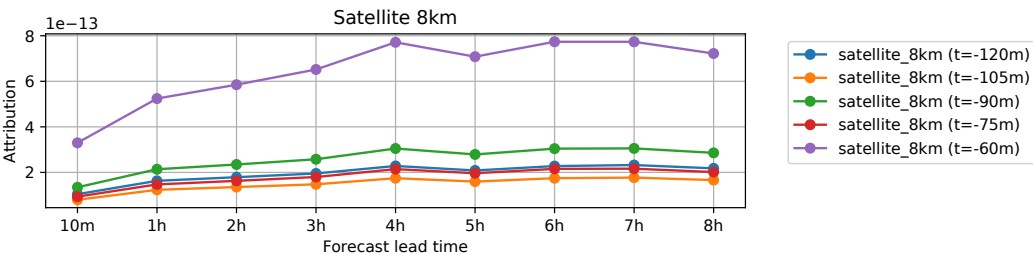

Figure 11: Attribution of satellite_8km input timesteps over forecast lead time.

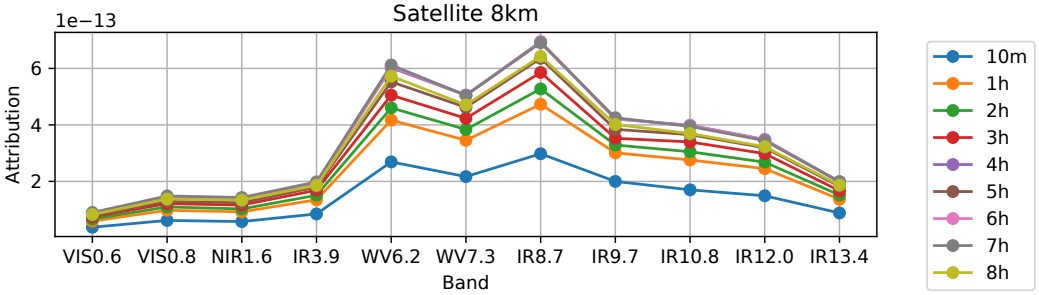

Figure 12: Attribution of satellite_8km input variables over forecast lead time.

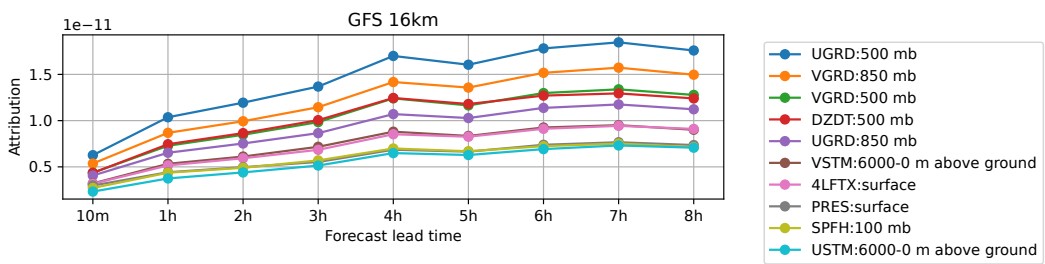

Figure 13: Attribution of gfs_16km input variables over forecast lead time.

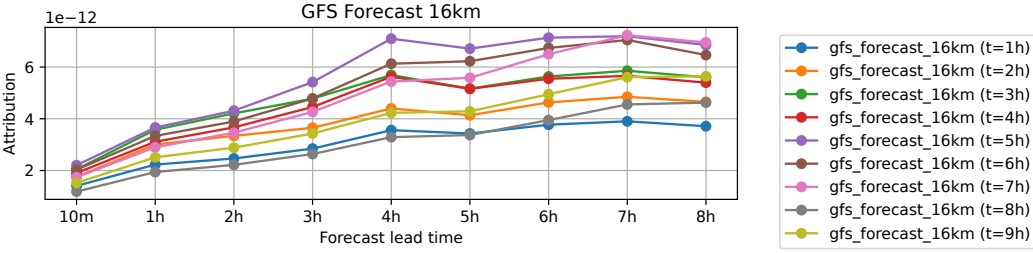

Figure 14: Attribution of gfs_forecast_16km input timesteps over forecast lead time.

# D   VISUALIZATIONS

This section displays a Europe-wide comparison of forecast models and probability maps for RainPro-8 at various precipitation levels, based on three different origins.

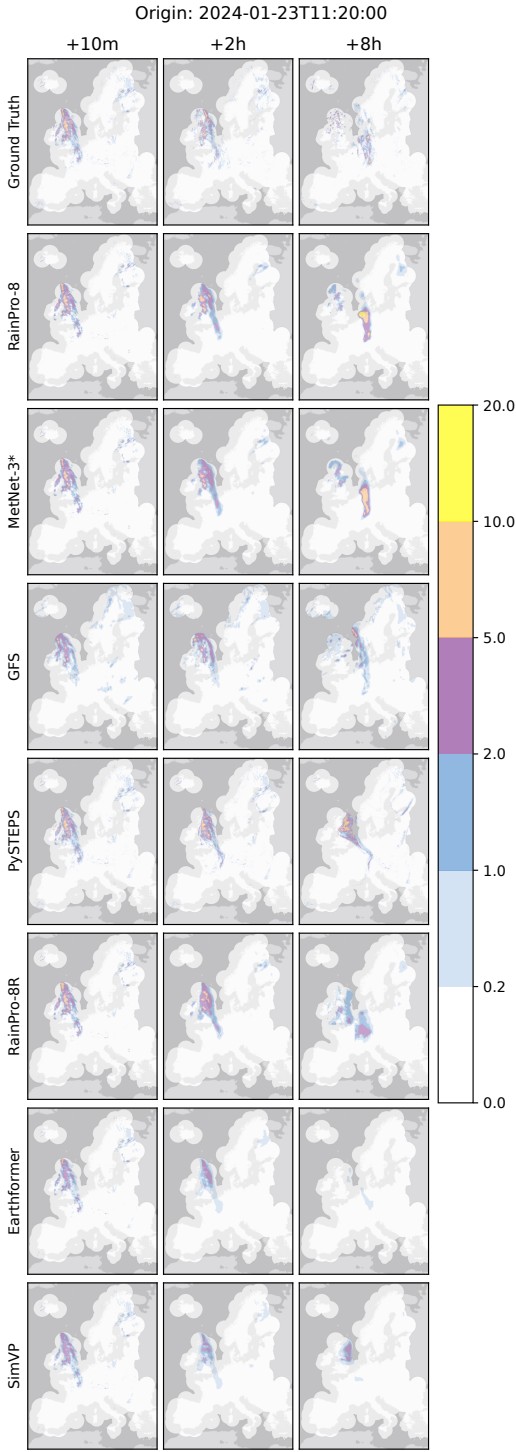

Figure 15: Sample ground truth, RainPro-8, MetNet-3*, GFS, and PySTEPS, RainPro-8R, Earth-former, and SimVP forecasts at selected lead times with origin at 2024-01-23 11:20 UTC.

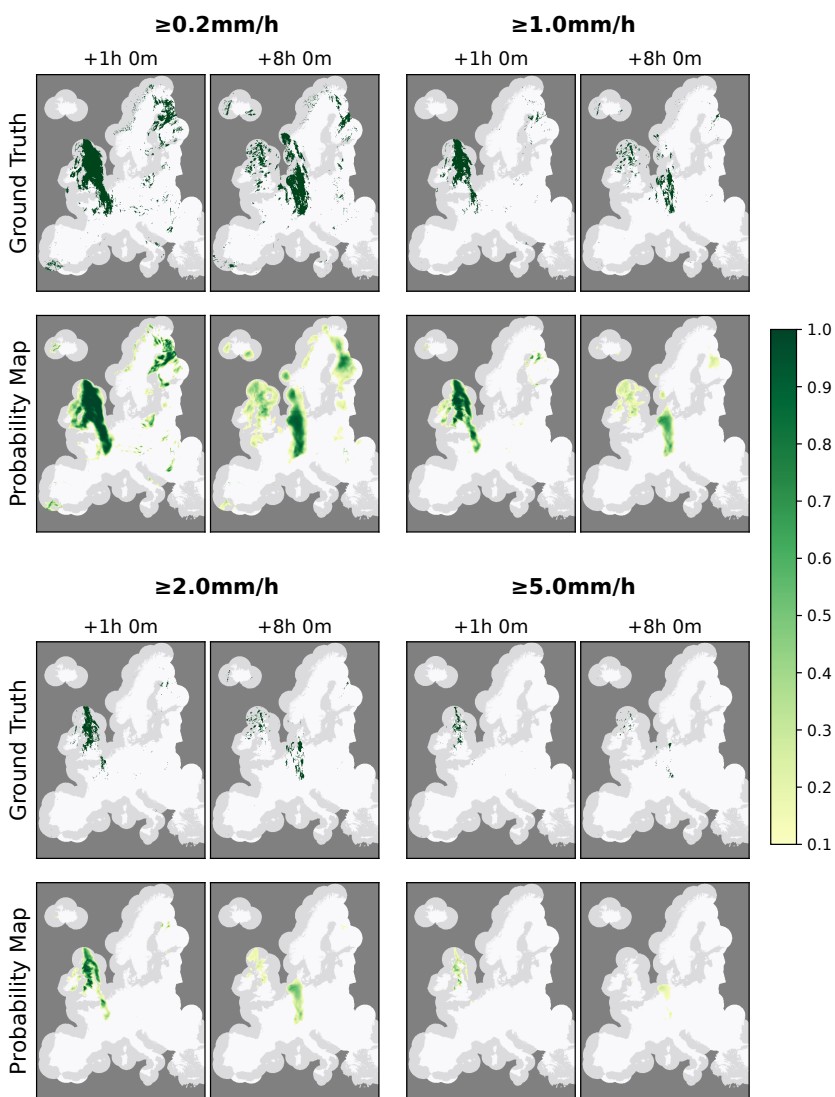

Figure 16: Probability map with corresponding ground truth for four different rain intensities and two different lead times with origin at 2024-01-23 11:20 UTC.

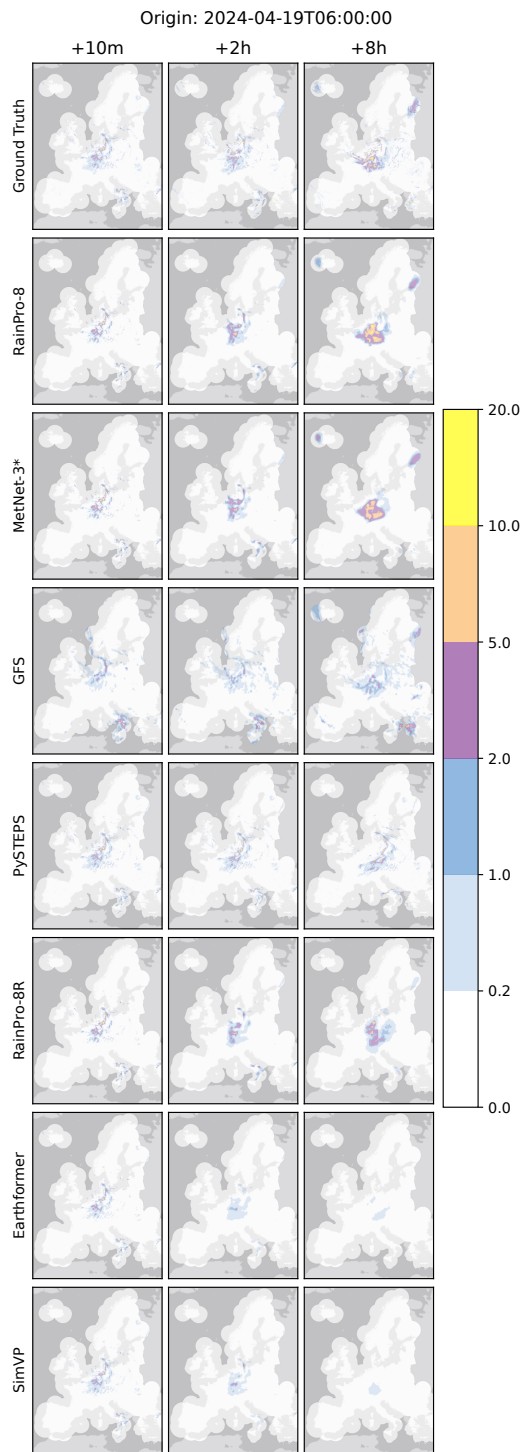

Figure 17: Sample ground truth, RainPro-8, MetNet-3*, GFS, and PySTEPS, RainPro-8R, Earthformer, and SimVP forecasts at selected lead times with origin at 2024-04-19 06:00 UTC.

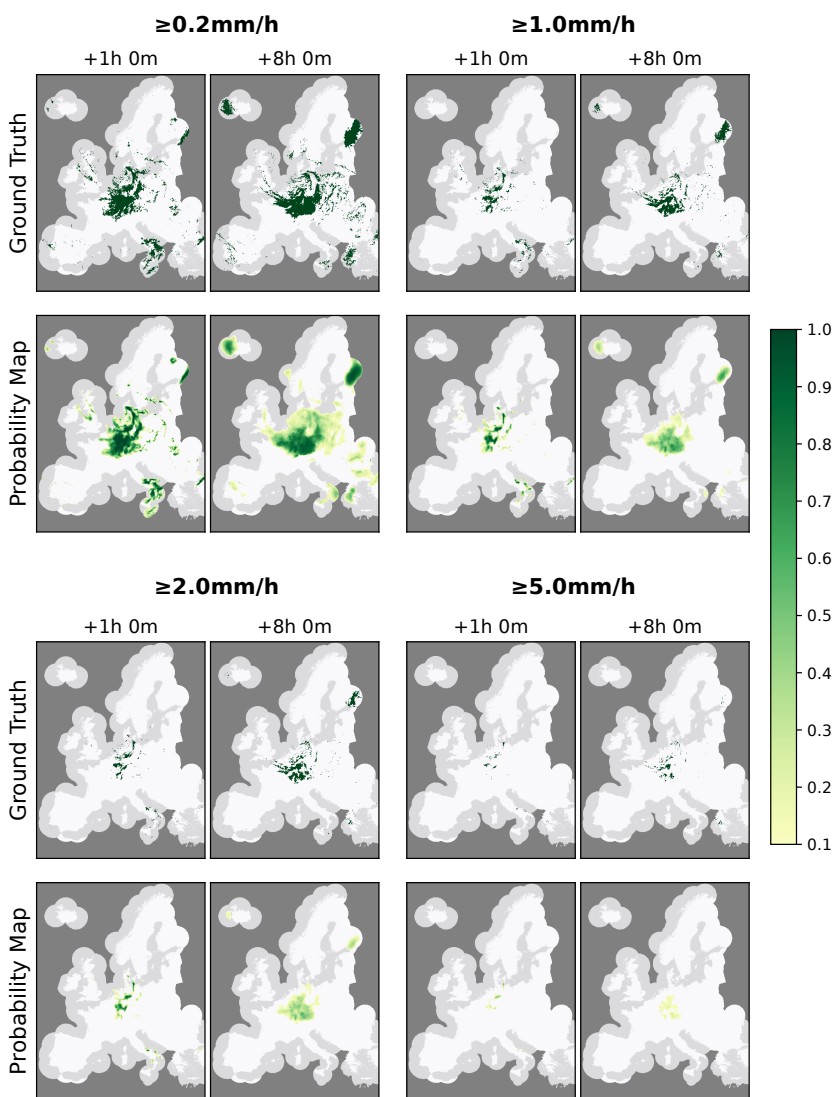

Figure 18: Probability map with corresponding ground truth for four different rain intensities and two different lead times with origin at 2024-04-19 06:00 UTC.

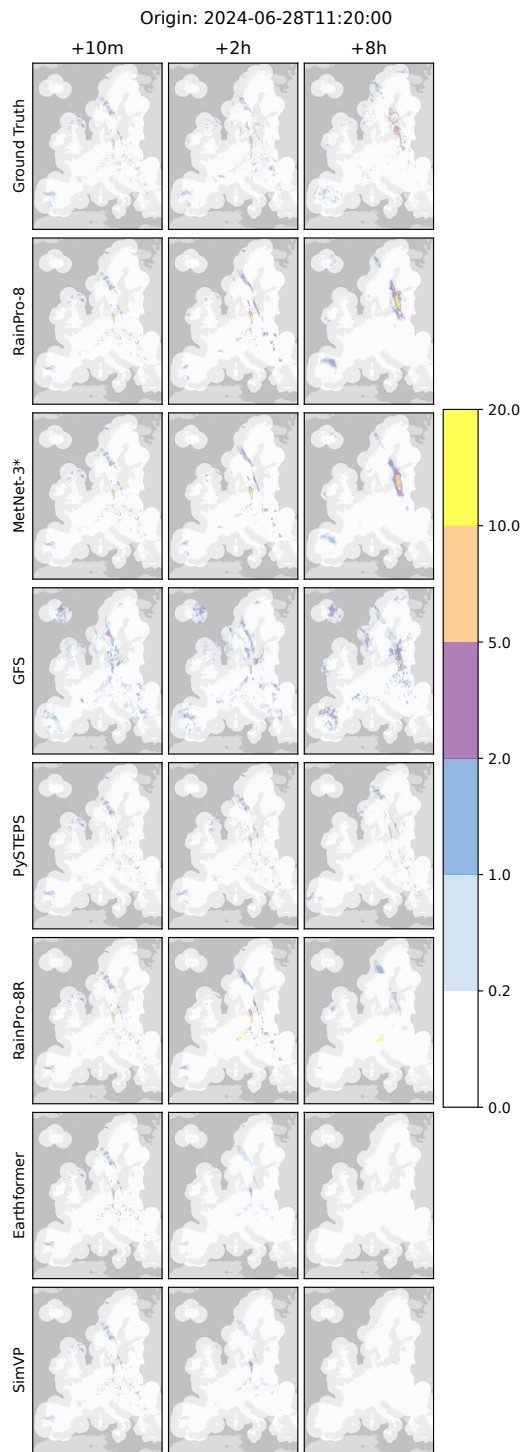

Figure 19: Sample ground truth, RainPro-8, MetNet-3*, GFS, and PySTEPS, RainPro-8R, Earthformer, and SimVP forecasts at selected lead times with origin at 2024-06-28 11:20 UTC.

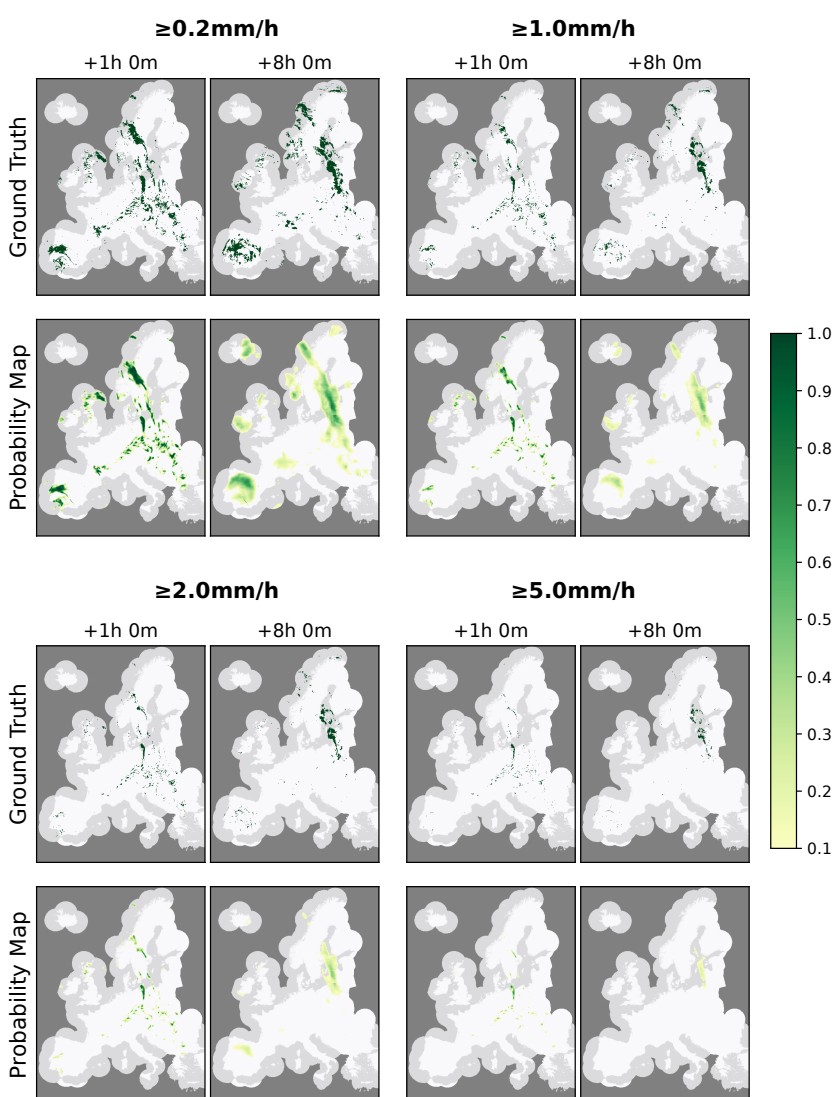

Figure 20: Probability map with corresponding ground truth for four different rain intensities and two different lead times with origin at 2024-06-28 11:20 UTC.

# E    INTENSITY CLASSES

Table 8 presents the intensity classes used by our model along with their distribution across different splits. Note that both the validation and test sets contain significantly more missing values due to patches with radar coverage below 50% to obtain Europe-wide forecasts.

Table 8: Intensity classes and distribution for each split.

| Bucket (mm/h) | Train (%) | Validation (%) | Test (%) |
|---|---|---|---|
| $[0.0, 0.1)$ | 79.64 | 57.44 | 57.05 |
| $[0.1, 0.2)$ | 1.44 | 1.00 | 1.10 |
| $[0.2, 0.4)$ | 2.25 | 1.62 | 1.87 |
| $[0.4, 0.6)$ | 0.84 | 0.62 | 0.74 |
| $[0.6, 0.8)$ | 0.51 | 0.37 | 0.44 |
| $[0.8, 1.0)$ | 0.50 | 0.36 | 0.44 |
| $[1.0, 2.0)$ | 0.84 | 0.63 | 0.73 |
| $[2.0, 3.0)$ | 0.38 | 0.30 | 0.35 |
| $[3.0, 4.0)$ | 0.19 | 0.15 | 0.17 |
| $[4.0, 5.0)$ | 0.17 | 0.13 | 0.14 |
| $[5.0, 6.0)$ | 0.06 | 0.05 | 0.05 |
| $[6.0, 7.0)$ | 0.05 | 0.04 | 0.04 |
| $[7.0, 8.0)$ | 0.04 | 0.04 | 0.03 |
| $[8.0, 9.0)$ | 0.03 | 0.02 | 0.02 |
| $[9.0, 10.0)$ | 0.02 | 0.02 | 0.02 |
| $[10.0, 15.0)$ | 0.03 | 0.03 | 0.02 |
| $[15.0, 20.0)$ | 0.02 | 0.02 | 0.01 |
| $[20.0, 25.0)$ | 0.01 | 0.01 | 0.01 |
| $\geq 25.0$ | 0.02 | 0.02 | 0.01 |
| missing | 12.97 | 37.15 | 36.78 |

## F  COMPUTE RESOURCES

Training is performed on an NVIDIA H100 80GB SXM5 GPU with 26 vCPUs and 225 GiB RAM. Table 9 shows the training and inference times for each experiment, including baselines and ablation studies.

Table 9: Training and inference times for all models trained.

| Experiment | Training Time (h) | Inference Time (s/sample) |
|---|---|---|
| RainPro-8 | 13:36 | 0.03 |
| MetNet-3* | 18:16 | 2.12 |
| RainPro-8R | 13:22 | 0.03 |
| Earthformer | 24:03 | 0.03 |
| SimVP | 6:36 | 0.12 |
| RainPro-8 w/ CE loss (no OC) | 15:52 | 0.03 |
| RainPro-8 w/ lead time conditioning | 18:06 | 2.10 |
| RainPro-8 w/o lead time weights | 13:21 | 0.03 |

## G  ROBUSTNESS ANALYSIS

The primary experiments in this work were conducted with a single training run due to the substantial computational resources required. While this setup provides a fair basis for comparison, the closeness of certain results motivates an analysis of the variability that may arise from different random seeds. Therefore, we conducted three independent runs for the two most comparable experiments: RainPro-8 and its cross-entropy variant, *RainPro-8 w/ CE loss (no OC)*. Table 10 reports the mean and standard deviation across these runs for multiple evaluation metrics. The results show that both models converge consistently, with RainPro-8 demonstrating a slight but consistent advantage across all metrics.

Table 10: Aggregated results from three independent runs (mean ± std). Best results are highlighted in bold.

| Metric | RainPro-8 | RainPro-8 w/ CE loss (no OC) |
|---|---|---|
| CRPS | **0.06096 ± 0.00004** | 0.06099 ± 0.00002 |
| CSI | **0.27923 ± 0.00054** | 0.27832 ± 0.00032 |
| FSS | **0.53663 ± 0.00119** | 0.53532 ± 0.00041 |
| FBI | **1.26048 ± 0.01224** | 1.26999 ± 0.00242 |
| MAE | **0.12593 ± 0.00102** | 0.12610 ± 0.00043 |
| MSE | **1.50463 ± 0.01819** | 1.52142 ± 0.01898 |

# H    SEVIR NOWCASTING EXPERIMENTS

## H.1    DATASET AND PREDICTION TASK

SEVIR (Veillette et al., 2020) is a widely used benchmark for precipitation nowcasting, containing 20,393 weather events of radar frame sequences of 4-hour duration covering $384 \times 384$ km$^2$ (1 km/pixel) at 5-minute intervals. Following Diffcast (Yu et al., 2024), we formulate the prediction task as forecasting 20 frames given 5 initial frames ($5 \rightarrow 20$), and the spatial resolution is downsampled to $128 \times 128$ pixels to reduce computational cost.

## H.2    RAINPRO-2R

We modify RainPro-8 to create RainPro-2R, a radar-only variant suitable for short-term nowcasting. RainPro-2R uses only radar frames as input, without additional data sources or spatial context with respect to the targets. The model predicts 20 frames at a downsampled resolution of $128 \times 128$ pixels, while retaining the same three downsampling layers used in RainPro-8. Training follows the Diffcast (Yu et al., 2024) pipeline, employing our ordinal consistent loss and single-step prediction approach. RainPro-2R is trained using the Adam optimizer with a learning rate of 0.0001 for a total of 200K iterations, and the model with the lowest validation loss is selected for evaluation.

The intensity classes that define the probability distribution are based on the SEVIR benchmark thresholds: [0.0, 16.0, 31.0, 59.0, 74.0, 100.0, 133.0, 160.0, 181.0, 219.0, 255.0]. Given the shorter lead times, the decay rate for the lead time weights is set to 2.

## H.3    EVALUATION

To assess the accuracy of precipitation nowcasting, we evaluate RainPro-2R against both deterministic and stochastic generative models. Deterministic baselines include SimVP (Gao et al., 2022) and Earthformer (Gao et al., 2024b), which employ a recurrent-free strategy to generate all frames simultaneously, as well as PhyDNet (Guen & Thome, 2020), MAU (Chang et al., 2021), and ConvGRU (Shi et al., 2017a), which generate frames sequentially using recurrent strategies. For state-of-the-art generative baselines, we include STRPM (Chang et al., 2022), MCVD(Voleti et al., 2022), and PreDiff (Gao et al., 2024a).

Evaluation metrics follow prior work (Yu et al., 2024) and include Critical Success Index (CSI), Heidke Skill Score (HSS), pooled CSI, and visual perception metrics LPIPS Zhang et al. (2018) and SSIM Wang et al. (2004). In addition, we introduce the previously discussed Fraction Skill Score (FSS) and Continuous Ranked Probability Score (CRPS).

Details on CSI, CRPS, and FSS are in Appendix B.1. Pooled CSI consists of computing CSI after applying max-pooling to the predinctions and targets. Therefore, pooled CSI only checks whether any pixel within a neighborhood is hit, ignoring how many pixels match.

The models are evaluated at thresholds of [16, 74, 133, 160, 181, 219] (Yu et al., 2024). Pooled CSI is computed over 4×4 and 16×16 neighborhoods, and CRPS over 5×5 and 17×17 neighborhoods, to have 2 or 8 pixels in each direction.

**Heidke Skill Score (HSS)**    HSS measures forecast accuracy while accounting for correct hits expected by chance. The values range between -0.5 and 1.0, with scores greater than 0 indicating skill by improving over a random forecast.

$$\mathrm{HSS} = \frac{2\left(\mathrm{TP} \cdot \mathrm{TN} - \mathrm{FN} \cdot \mathrm{FP}\right)}{(\mathrm{TP} + \mathrm{FN})(\mathrm{FN} + \mathrm{TN}) + (\mathrm{TP} + \mathrm{FP})(\mathrm{FP} + \mathrm{TN})}, \tag{14}$$

where TP, FP, and FN are true positives, false positives, and false negatives, based on the threshold (rate $\geq$ threshold).

**Learned Perceptual Image Patch Similarity (LPIPS)**    LPIPS (Zhang et al., 2018) measures perceptual similarity between predicted and observed images using deep network features. It can cap-

ture differences in texture, structure, and high-level perceptual qualities. Lower LPIPS values indicate predictions that are perceptually closer to the observations.

**Structural Similarity Index Measure (SSIM)**    SSIM (Wang et al., 2004) evaluates image similarity by comparing luminance, contrast, and structural information between predicted and observed images. It reflects how well the predicted patterns match the true spatial structures. Higher SSIM values indicate better preservation of structural and visual information in the forecast.

# I GFS INPUT VARIABLES

Table 11: Selected GFS variables as input to the model with their corresponding levels.

| Var. | Description | Levels |
|---|---|---|
| 4LFTX | Best (4 layer) Lifted Index | surface |
| ABSV | Absolute Vorticity | 100mb, 250mb, 500mb, 850mb, 1000mb |
| CAPE | Convective Available Potential Energy | surface, 180-0mb, 90-0mb, 255-0mb |
| CFRZR | Categorical Freezing Rain | surface |
| CICEP | Categorical Ice Pellets | surface |
| CIN | Convective Inhibition | surface, 180-0mb, 90-0mb, 255-0mb |
| CLMR | Cloud Mixing Ratio | 250mb, 500mb, 850mb, 1000mb |
| CPOFP | Percent frozen precipitation | surface |
| CRAIN | Categorical Rain | surface |
| CSNOW | Categorical Snow | surface |
| CWAT | Cloud Water | entire atmosphere |
| DPT | Dew Point Temperature | 2m |
| DZDT | Vertical Velocity (Geometric) | 100mb, 250mb, 500mb, 850mb, 1000mb |
| GRLE | Graupel | 100mb, 250mb, 500mb, 850mb, 1000mb |
| GUST | Wind Speed (Gust) | surface |
| HGT | Geopotential Height | 100mb, 250mb, 500mb, 850mb, 1000mb, surface, trop. |
| HPBL | Planetary Boundary Layer Height | surface |
| ICEG | Ice Growth Rate | 10m |
| ICETK | Ice Thickness | surface |
| ICMR | Ice Water Mixing Ratio | 250mb, 500mb, 850mb, 1000mb |
| LCDC | Low Cloud Cover | low cloud layer |
| LFTX | Surface Lifted Index | surface |
| MCDC | Medium Cloud Cover | middle cloud layer |
| MSLET | (Eta model reduction) | mean sea level |
| PLPL | Pressure of level from which parcel was lifted | 255-0mb |
| PRATE | Precipitation Rate | surface |
| PRES | Pressure | surface, trop. |
| PRMSL | Pressure Reduced to MSL | mean sea level |
| PWAT | Precipitable Water | entire atmosphere |
| REFC | Composite reflectivity | entire atmosphere |
| RH | Relative Humidity | 100mb, 250mb, 500mb, 850mb, 1000mb, 2m, entire atmosphere |
| RWMR | Rain Mixing Ratio | 100mb, 250mb, 500mb, 850mb, 1000mb |
| SNMR | Snow Mixing Ratio | 100mb, 250mb, 500mb, 850mb, 1000mb |
| SNOD | Snow Depth | surface |
| SPFH | Specific Humidity | 100mb, 250mb, 500mb, 850mb, 1000mb, 2m |
| TCDC | Total Cloud Cover | 250mb, 500mb, 850mb, 1000mb, entire atmosphere |
| TMP | Temperature | 100mb, 250mb, 500mb, 850mb, 1000mb, surface, 2m, trop. |
| UGRD | U-Component of Wind | 100mb, 250mb, 500mb, 850mb, 1000mb, 10m, trop. |
| USTM | U-Component Storm Motion | 6000-0m |
| VGRD | V-Component of Wind | 100mb, 250mb, 500mb, 850mb, 1000mb, 10m, trop. |
| VIS | Visibility | surface |
| VSTM | V-Component Storm Motion | 6000-0m |
| VVEL | Vertical Velocity (Pressure) | 100mb, 250mb, 500mb, 850mb, 1000mb |
| VWSH | Vertical Speed Shear | trop. |
| WEASD | Water Equivalent of Accumulated Snow Depth | surface |

## J LLM USAGE

Large Language Models have been employed solely for proofreading and to polish writing.

