# OpenReview forum: "RainPro-8: An Efficient Deep Learning Model to Estimate Rainfall Probabilities Over 8 Hours"
_ICLR.cc/2026/Conference — ICLR 2026 Poster_

### Official Review · Reviewer_MSzr · 2025-10-21

**Soundness:** 3
**Presentation:** 3
**Contribution:** 3
**Rating:** 6
**Confidence:** 4

**Summary:**

The paper introduces RainPro-8, a deep-learning model for probabilistic precipitation forecasting. It proposes an Ordinal-Consistent Loss that maintains the natural ordering of rainfall intensity classes and a Single-Pass Prediction method that predicts all future timesteps at once for greater efficiency and temporal consistency. Built on a U-Net with MaxViT blocks, the architecture efficiently processes multi-resolution atmospheric inputs with significantly fewer parameters than prior MetNet models.

**Strengths:**

- The paper identifies the limitations of radar-only forecasting methods and proposes a multi-source approach to improve prediction accuracy.
- It provides a detailed analysis of recent Climate AI research, highlighting their characteristics and limitations, and conducts experiments following established protocols and evaluation metrics.
- To address the ordinal nature of precipitation classes and the difficulty of long lead-time forecasts, it introduces a specialized loss function and a single-pass prediction method.
- The appendix provides comprehensive details on the variables, data assimilation, and preprocessing steps for the diverse data sources used.

**Weaknesses:**

- Including the baseline model MetNet, RainPro-8 relies on a variety of data sources, which limits its applicability to regions—such as developed countries—where high-resolution data from multiple sensors, satellites, and NWP outputs are readily available.

- The model lacks significant technical novelty. Moreover, its performance improvement over the existing baseline, MetNet-3, is marginal.

**Questions:**

- Many models that perform 6-hour nowcasting are discussed in the Related Works section. What is the logical reasoning and supporting justification for forecasting 8 hours instead of 6, as done in those prior works?
- Besides SEVIR, is it possible to evaluate performance on other relevant benchmark datasets such as the Shanghai dataset or CIKM?
- In Line 109, the paper states that “its training requires significant time and resources, involving hundreds of Tensor Processing Units (TPUs) for multiple days.” However, if this method uses an “NVIDIA H100 SXM5 GPU,” isn’t that also an extremely powerful computational setup? Please summarize the computational power and training duration for the compared baseline methods.

**Details Of Ethics Concerns:**

Since the model utilizes weather and precipitation data, there are no relevant ethics concerns.

---

> ### Author Response · Authors · 2025-11-18
>
> We thank Reviewer MSzr for their feedback and for recognizing the strengths of our multi-source approach, methodological contributions, thorough analysis of related work, and comprehensive experimental setup. Below, we address the concerns and questions raised.
>
> **Weakness 1 – Applicability to high-resource regions only**
> High-resolution precipitation forecasting often relies on radar data, which is not available everywhere, while global satellite and NWP data are more widely accessible. This limitation applies to baseline models as well, including MetNet and other radar-based nowcasting models using SEVIR, Shanghai, or CIKM datasets. As noted in Future Work, improving robustness to missing data is an ongoing direction. In regions without radar, satellite-derived products such as IMERG GPM could be used, though this is beyond the scope of the current work.
>
> **Weakness 2 – Novelty and performance vs. MetNet-3**
> For the first time, RainPro-8 offers a computationally effective, efficient, and open high-resolution forecasting model, making a strong contribution to resource-aware deep learning models in this applications to physical sciences track. We demonstrate strong empirical performance on a high-impact, under-served problem. The model is compact, reproducible, and effective for 8-hour probabilistic precipitation forecasting, achieving x48 faster inference and consistent probabilistic outputs thanks to the single-pass prediction and ordinal-consistent loss. RainPro-8’s learning and architectural advances offer clear impact in a field where leading models are often closed and extremely resource intensive. MetNet-3 lacks publicly available code and data, is highly resource-intensive, and depends on exclusive U.S. datasets, making independent reproduction challenging.
>
> **Question 1 – 6-hour forecasting models**
> We chose 8-hour forecasting to support daily operational workflows, providing accurate predictions for the full workday. The model is easily adaptable to shorter lead times, such as 2 hours (RainPro-2R), or any desired horizon. We did not directly compare to the 6-hour models cited in Related Work because they either target different tasks (e.g., radar-free forecasting), rely on unavailable data or code, or evaluate at hourly resolution or on rainy days. Instead, we compare RainPro-8 with operational products (PySTEPS, GFS), state-of-the-art forecasting models (MetNet-3*), and top deep learning nowcasting methods (Earthformer, SimVP), covering methods whose strengths span from the early hours to multi-day forecasting scales.
>
> **Question 2 – Short-term precipitation nowcasting benchmarks**
> We used the SEVIR benchmark to address potential dataset concerns, namely, very short time horizons. Our primary task are 8-hour forecasts, where multiple data sources, efficient high-resolution outputs, and uncertainty are critical. Nevertheless, we will explore additional benchmark evaluations in the Appendix for completeness, even though such sources are only available for 2 hours (Shanghai) and 1 hour (CIKM).
>
> **Question 3 – Compute clarification for MetNet-3 vs. our model**
> RainPro-8 was trained on a single NVIDIA H100 SXM5 GPU for 13 hours (approximately 45 USD on LambdaLabs), compared with MetNet-3’s reported training setup of 512 TPUv3 cores for 7 days. Using standard TPUv3 pricing of 2.00 USD per hour per chip, with each chip containing 2 cores, this corresponds to 256 chips at 512 USD per hour. The total estimated cost is therefore 7 * 24 * 512 USD = 86,016 USD, whereas RainPro-8’s training cost is only 45 USD. This means RainPro-8 is roughly x1900 cheaper to train than MetNet-3. On smaller GPUs such as the NVIDIA L40S 48GB, training can be completed in under 24 hours. Regardless of hardware, the model remains compact (20% the size of MetNet-3) and provides x48 faster inference due to its single-pass prediction design.

---

### Official Review · Reviewer_mScv · 2025-10-24

**Soundness:** 3
**Presentation:** 2
**Contribution:** 2
**Rating:** 6
**Confidence:** 4

**Summary:**

This paper presents RainPro-8, an efficient and compact deep learning model for 8-hour high-resolution probabilistic precipitation forecasting in Europe. RainPro-8 integrates heterogeneous multi-source data (radar, satellite, and numerical weather prediction, NWP), and adopts an optimized U-Net + MaxViT architecture. The model generates probabilistic maps for all time steps and precipitation classes in a single forward pass. The authors introduce an "ordinal consistent loss" to explicitly encode the ordinal nature of precipitation bins, improving interpretability and consistency of the probabilities. RainPro-8 uses less than 20% of MetNet-3's parameters but achieves superior performance across lead times and precipitation intensities, outperforming NWP baselines, extrapolation methods, and the latest deep learning models (e.g., Earthformer, SimVP, MetNet-3*). Extensive experiments, including ablations, attribution, and efficiency analysis, are provided, as well as competitive results on the short-term SEVIR benchmark. Code will be public, and datasets and experimental details are thorough.

**Strengths:**

RainPro-8 efficiently combines heterogeneous multi-source data, achieves multi-step probabilistic forecasting in one pass with much fewer parameters than strong baselines, and its ordinal consistent loss is specifically designed for the nature of precipitation bins. The model consistently outperforms state-of-the-art competitors on accuracy, efficiency, and uncertainty quantification.

**Weaknesses:**

The “ordinal consistent loss” (Section 3.2) is a core novelty, but its theoretical justification is limited. The paper gives only a high-level description and refers to another field (semantic segmentation) for motivation. There is no theoretical or empirical exploration of why ordinal modeling is essential for probabilistic precipitation. Ablation is very basic, and critical aspects such as loss stability, parameter sensitivity, and generalization are not deeply examined.

While RainPro-8 claims to handle varying precipitation intensities, most experiments focus on average metrics. The model’s ability to capture extreme/rare strong precipitation events (e.g., ≥25 mm/h) is not thoroughly assessed. Only the class coverage is reported, but there are no case studies or targeted metrics for extreme events, missing a rigorous quantification of robustness or uncertainty calibration for outlier scenarios.

**Questions:**

does the ordinal consistent loss have any theoretical advantage (e.g., calibration, capturing uncertainty, or predicting extremes)? Could you provide more empirical or theoretical analysis on loss stability and generalization?

In known European extreme rainfall cases or strong out-of-distribution samples, how does RainPro-8's predicted probability align with actual precipitation? Are confidence intervals and rare event probabilities calibrated and unbiased?

---

> ### Author Response · Authors · 2025-11-18
>
> We thank Reviewer mScv for their feedback and for recognizing the overall strengths of our approach, including its modeling design, efficiency, and consistent performance improvements. Below, we address the concerns and questions raised.
>
> **Weakness 1 – Ordinal Consistent loss details**
> Cross-entropy loss (MetNet) treats each precipitation class independently - predictions are simply right or wrong. We use ordinal consistent loss to penalize mispredictions based on class ordering: predicting no rain when it rains heavily should be worse than predicting light rain. This requires modeling accumulated probabilities ($P(R ≥ \textrm{threshold})$) with conditional probabilities as outputs, which mathematically enforces monotonicity: $P_{t,c} ≤ P_{t,c-1}$ by design. This prevents non-monotonic outputs like predicting higher probability for 10 mm/h than 1 mm/h. Appendix G validates training stability across three independent runs with negligible variance, demonstrating robust convergence. We will revise our motivation to include this discussion.
>
> **Weakness 2 – Strong precipitation**
> For strong precipitation (≥10 mm/h, classified as heavy showers by the WMO), RainPro-8 demonstrates robust calibration using intensity-specific metrics such as CSI, FSS, and FBI at the 10mm/h threshold (Tables 5-7). CSI, also known as Threat Score, provides a balanced assessment, as it accounts for both false alarms and missed events, unlike POD or FAR alone.  Thus, CSI is a metric that captures RainPro-8's ability to detect extreme events. FSS similarly reflects detection skill while tolerating small spatial shifts and transformations, and FBI measures how well predicted event frequencies match observations, showing reasonable calibration despite <0.1% of training samples falling in this range.
>
> **Question 1 – Ordinal Consistent loss advantages**
> The ordinal formulation ensures calibration by eliminating internally inconsistent probability distributions through monotonicity constraints. It enables better uncertainty quantification since the hierarchical conditional probability structure naturally captures cascading uncertainty (predicting extreme rainfalls requires first being confident about lighter precipitation). For predicting extremes, the masked loss allocates full gradient capacity to rare events when they occur, preventing gradient dilution across the majority no-rain class. We will highlight this more clearly in the final version. Regarding stability and generalization, Appendix G shows that training across three independent runs yields negligible variance, demonstrating consistently robust convergence.
>
> **Question 2 – Extreme rainfall probabilities**
> Although we did not explicitly examine recent extreme events, the dataset spans diverse high-intensity precipitation cases across Europe in 2024. CSI at 10 mm/h (Table 5) directly measures skill on actual extreme events while ignoring the trivial no-rain class, showing strong performance despite the rarity of these events. The FBI of 1.636 at 10 mm/h (Table 7) demonstrates that the model assigns meaningful probability to extremes and accurately captures their observed frequencies. Monotonic probability maps (Figure 4) ensure that predictions of heavy rain maintain consistent orderings across intensities, making the forecasts reliable for operational applications.

---

> > ### Comment · Reviewer_mScv · 2025-11-20
> >
> > Thank you for the authors' response. I believe a score of 6 is appropriate for this paper, and I will therefore maintain my rating.

---

### Official Review · Reviewer_iBcq · 2025-10-31

**Soundness:** 3
**Presentation:** 2
**Contribution:** 2
**Rating:** 4
**Confidence:** 4

**Summary:**

The paper describes a  deep-learning model for high-resolution probabilistic precipitation
forecasting over an 8-hour horizon. The main advantage of the proposed model is in its ability to predict precipitation for a longer time frame than the standard nowcasting, up to 2 hours. To achieve this goal, the authors fuse radar, satellite, and physics-based nu-
merical weather prediction (NWP) data.

**Strengths:**

The paper covers an important topic, precipitation forecasting.

Originality: the paper leverages multi-source prediction and claims prediction of up to 8 hours (however, see the weaknesses below)

Quality: the paper is well-written in general (however, there are caveats as some parts of the paper are difficult to follow). The equations, as I checked, are correct

**Weaknesses:**

Clarity and quality: see questions below

Significance: I think the authors need to clarify on this point.  The contributions cite the following:
-  *Efficient architecture and training strategy*: however, the architecture seems to be a well-parameterised UNet-based model (Figure 1). The authors state: "Key differences include single-pass predictions without lead time conditioning (Section 3.3), early downsampling in the encoder, halving internal channels, and removing topographical embeddings, all contributing to a reduced parameter count of 36.7M from the original 227M." Would this be the contribution? I think it could be better to have some sort of takeaway message justifying these architectural solutions, and why it could help develop better new precipitation forecasting architectures
- "significant gains over deep-learning nowcasting models" See Q1
- "Demonstration of RainPro’s versatility for radar-only 2-hour predictions on the SEVIR benchmark, achieving state-of-the-art performance compared to both deterministic and generative nowcasting models" I am not entirely sure I get it. If the claim is to get 8 hour prediction, is achieving state-of-the-art results on 2-hour radar predictions of the contribution?

**Questions:**

1. "Extensive empirical evaluation demonstrating that RainPro-8 outperforms existing operational methods by 65% " I am not sure I can follow where this is described, it only appears in the introduction
2. I am not sure I can follow Figure 2, it is very small. From what I can follow, does the proposed method essentially follow very closely MetNet-3*, is that correct? In that case, the main claim could be that you achieve slightly higher performance, but reduce the computational costs  48 times.
3. For Table 3, the ablation study results show only small differences in performance. Could the authors give confidence intervals, if possible?

---

> ### Author Response · Authors · 2025-11-18
>
> We thank Reviewer iBcq for their feedback and for recognizing the relevance of the problem, the general clarity and quality of the work, and the value of exploring multi-source forecasting. Below, we address the concerns and questions raised.
>
> **Weakness 1 – Clarity and Quality:** questions addressed below
>
> **Weakness 2 – Significance**
> * The architectural modifications (early downsampling, halving internal channels, and removing topographical embeddings) mainly reduce the parameter count from 227M to 36.7M, creating a more efficient model. To further enhance both accuracy and efficiency, we propose single-pass predictions with lead-time weighting and the ordinal-consistent loss. Combined, these design and training strategies provide a practical framework for developing high-performance, resource-efficient precipitation forecasting models. We will clarify this first contribution in the final version to highlight its significance.
> * Q1 addressed below
> * RainPro-8 is designed for 8-hour precipitation forecasting, using a UNet with MaxVit blocks to fuse multiple data sources, together with single-pass predictions and the ordinal-consistent loss to improve efficiency and predictive accuracy. The SEVIR radar-only 2-hour evaluation uses a simpler architecture with only radar but applies the same training and architectural strategies, demonstrating that the ordinal consistent loss to generate consistent probabilistic outputs, lead-time weighted single-pass predictions, and efficient design generalize beyond the full 8-hour, multi-source setup and underline the strengths of RainPro-8.
>
> **Question 1 – Improvement calculation**
> The 65% improvement is a conservative estimate based on comparing RainPro-8 (mean CSI = 0.279) with an “optimal operational baseline” (mean CSI = 0.169), obtained by taking, at each threshold and lead time, the maximum CSI of PySTEPS (better at short lead times) and GFS (better at long lead times). This baseline performs better than PySTEPS alone (mean CSI = 0.149) and GFS alone (mean CSI = 0.110). We will clarify this in the final version.
>
> **Question 2 – Figure 2 and improvements**
> In the final version, we will enlarge Figure 2. Although its visible differences are subtle in Figure 2, Tables 2, 5, 6, and 7 show clearer gaps across aggregated metrics and thresholds. Additionally, it indeed provides a x48 inference-time speedup through single-pass prediction.
>
> **Question 3 – Confidence**
> To confirm that RainPro-8 indeed shows superior performance, we provide Appendix G, with three independent runs for the two main experiments, in the interest of computational cost. The aggregated results from these runs demonstrate that both models converge consistently, with RainPro-8 consistently outperforming competitors. Each run costs approximately 45 USD, so extending the full ablation study to three seeds would require roughly an additional 500 USD in cloud compute.

---

> > ### Comment · Reviewer_iBcq · 2025-11-27
> >
> > After checking the responses to my comments, as well as  Reviewer AAd8, I decided to retain my score. I think that the main concern is about the contribution towards the representation learning aspects. This work proposes improved training strategy, but does not necessarily propose new representation learning techniques.
> >
> > The concerns about baselines are another open question; without comparing with generative-modelling precipitation forecasting baselines in the meteorological domain.
> >
> > Finally, in the results with the discussion with Reviewer AAd8 there are concerns raised about long-term over-forecasting problem (FBI Score).

---

> > > ### Author Response · Authors · 2025-11-27
> > >
> > > We thank the reviewer for their engagement and for elaborating on the remaining concerns.
> > >
> > > **Contributions**
> > > While RainPro-8 builds on the MetNet-3 architecture, its contributions go beyond an improved training strategy. It provides a **unified, single-pass, multi–lead-time probabilistic** forecasting framework that integrates **multiple data sources** with a **precipitation-specific training objective**. For the **Applications to Physical Sciences track**, we believe the combination of empirical performance, accessibility, and domain impact aligns with the track’s goals, even if some representation-learning techniques are incremental.
> > >
> > > **Baselines and generative models**
> > > As discussed in our response to Reviewer AAd8, the **SEVIR evaluation** already includes comparisons with **generative models**. **NowcastNet and DGMR focus on short-term radar-only nowcasting**, and because their code is not public, key implementation details cannot be reproduced, making any re-implementation inherently unreliable and not a fair baseline. RainPro-2R, built on the same framework, architecture, and loss, already outperforms both deterministic and generative baselines studied in DiffCast (CVPR 2025). We remain open to evaluating any feasible open-source models the reviewer recommends.
> > >
> > > **Performance evaluation**
> > > We refer to our response to Reviewer AAd8 for detailed metrics. In short, **RainPro-8 provides accurate high-resolution forecasts up to 8 hours**. Despite a slight overprediction at the final timestep, this is not applicable in most timesteps (Figure 8), and **skill metrics (CSI & FSS) that penalize both false negatives and false positives show better performance for RainPro-8**.
> > >
> > > We hope these clarifications, together with our response to Reviewer AAd8, address the reviewer’s remaining concerns and are taken into account for the final decision.

---

### Official Review · Reviewer_AAd8 · 2025-11-01

**Soundness:** 2
**Presentation:** 2
**Contribution:** 1
**Rating:** 2
**Confidence:** 4

**Summary:**

The authors propose RainPro-8, a deep learning model for high-resolution, 8-hour probabilistic precipitation forecasting over Europe. The model is based on the MetNet-3 architecture but is modified to have significantly fewer parameters. It introduces a *single-pass prediction* strategy to improve inference efficiency and an Ordinal Consistent Loss to handle probabilistic bins. The authors claim the model surpasses existing deep learning and numerical weather prediction methods on several metrics.

**Strengths:**

1. The paper addresses the challenge of 8-hour, high-resolution probabilistic precipitation forecasting. This is a critical and difficult task that bridges the gap between traditional nowcasting and medium-range forecasting.
2. The usage of Ordinal Consistent Loss models the conditional probability of exceeding intensity thresholds, is designed to explicitly account for the ordinal structure of precipitation classes , which is a more principled approach than using a standard cross-entropy loss that treats classes as independent.

**Weaknesses:**

1. **Limited Methodological Novelty (*Main Weakness*)**: The primary weakness of this paper lies in its limited methodological novelty relative to ICLR standards, which emphasize fundamental advances in learning representations.
- Aside from the new loss function, the work's primary contribution is an application of existing techniques to create an efficient system.
- The model architecture is just based on MetNet-3. The main architectural changes include *early downsampling* and *halving internal channels*. These are standard engineering practices for model compression and efficiency, not **novel architectural designs or new methods for learning representations**.
2. **Lack of comparison on GAN-based models**: The paper's experimental validation lacks a crucial component in its discussion of *GAN-based generative models for precipitation*. In the related work section (Section 2), the authors correctly identify the importance of deep generative models, citing high-impact work such as DGMR (Ravuri et al., 2021) and NowcastNet (Zhang et al., 2023). These models, both published in *Nature* and recognized for their strong performance, are a key pillar of the state of the art in this field.
Despite acknowledging this work, the main experimental comparison in Table 1 completely **omits comparisons against these (or any other) GAN-based generative methods**. This is a significant gap in the evaluation. Without benchmarking against this entire class of SOTA models, it is impossible for the reader to assess the paper's true performance, and the claim to "surpass... deep-learning nowcasting models"  remains unsubstantiated.
3. **Contradiction Between Quantitative Metrics and Qualitative Results**: A significant weakness undermines the paper's experimental conclusions: the quantitative metrics and the qualitative case studies appear to contradict each other directly.
- Quantitative Metrics: The paper's aggregated metrics, specifically the Frequency Bias Index (FBI) in Table 7 and Figure 8, suggest a key advantage for RainPro-8 in forecasting heavy precipitation. At the 10.0 mm/h threshold, RainPro-8 reports an FBI of 1.636, which is notably lower (i.e., less over-forecasting) than the 1.821 reported for MetNet-3*. **This metric suggests the author's model is more balanced**.
- Case Study Results: The paper's own qualitative visualizations repeatedly show the opposite. In the case study for the +8h forecast in Figure 19, the Ground Truth shows only a small area of precipitation at the >10.0 mm/h (yellow) level. However, **RainPro-8 predicts a large, distinct area of heavy rain, while the MetNet-3 forecast for the same event shows a substantially smaller overforecasted area**. This same pattern, where RainPro-8 visually over-forecasts heavy rain more severely than MetNet-3, is apparent in the other examples provided (Figure 17).
This fundamental discrepancy, in which all qualitative examples in the paper contradict the aggregate metrics for heavy rain, is neither acknowledged nor explained. This casts serious doubt on the reliability of the reported results and the validity of the paper's evaluation.

**Questions:**

See weaknesses.

---

> ### Author Response · Authors · 2025-11-18
>
> We thank Reviewer AAd8 for their feedback and for recognizing the importance and difficulty of high-resolution, medium-range precipitation forecasting, as well as the principled design of our loss to account for the ordinal structure of precipitation classes. Below, we address the concerns raised.
>
> **Weakness 1 – Methodological novelty**
> RainPro-8 main contributions are not so much in the architecture where we mainly reduce parameter count, but rather in the training and inference innovations: the ordinal-consistent loss ensures coherent probabilistic outputs across precipitation classes, and single-pass predictions with lead-time weighting improve both accuracy and inference efficiency. This design is well-motivated to account for class ordinality and efficiently generate forecasts, and validated through extensive experiments, comparing also with the MetNet-3* setup, and thorough ablations. Importantly, RainPro-8 provides a practical, open-source model that can be trained on a single GPU, unlike MetNet-3, making high-resolution 8-hour probabilistic forecasting accessible. MetNet-3 lacks public access to its code and data and is extremely computationally expensive. RainPro-8 model and training choices, as well as the operational efficiency, represent a meaningful contribution to the field, in particular in this applications to physical sciences track.
>
> **Weakness 2 – GAN-based models**
> While comparisons with GAN-based models could potentially further contextualize performance, they are actually not state-of-the-art for our tasks. Rather, models such as NowcastNet and DGMR do short-term radar-only nowcasting, similar to the generative models evaluated in Section 4.5, limited in forecast skill and horizon due to lack of additional sources and uncertainty handling. In addition, their code is not publicly available, highlighting a key motivation for RainPro-8: providing a compact, transparent, and reproducible solution for state-of-the-art high-resolution precipitation forecasting. To assess performance on equal terms despite these challenges, our empirical evaluation also includes generative models on radar-only SEVIR on their short 2-hour prediction task in Section 4.5, where RainPro-2R outperforms generative models in CSI, HSS, CRPS, and FSS. Importantly, results confirm that radar-only approaches are insufficient for accurate 8-hour forecasting.
>
> **Weakness 3 – Quantitative metrics and qualitative results**
> Thank you for asking this important question: actually, these qualitative images are not representative, and while in some cases, including some of the ones shown, there is some degree of overprediction, this is not true overall, as is accurately reflected in our quantitative metrics. Table 7 and Figure 8 show that RainPro-8 is overall more balanced than MetNet-3* (FBI 1.636 vs. 1.821 at 10 mm/h), indicating that its predicted event frequency is closer to observations across most lead times. As shown in Figure 8, it is true that at the 8h timestep, RainPro-8 slightly overpredicts compared to MetNet-3* (FBI 2.220 vs. 2.157). However, this is not representative of other timesteps. To further support the reliability of our results, we provide a table of total positive observations (%), positive predictions by RainPro-8 and MetNet-3*, and their corresponding FBI every 30 minutes, confirming the validity of the evaluation.
>
> | Lead Time (h) | 0.5 | 1.0 | 1.5 | 2.0 | 2.5 | 3.0 | 3.5 | 4.0 | 4.5 | 5.0 | 5.5 | 6.0 | 6.5 | 7.0 | 7.5 | 8.0 |
> |-----------------------------------|-----|-----|-----|-----|-----|-----|-----|-----|-----|-----|-----|-----|-----|-----|-----|-----|
> | Total Positive Observations (%) | 0.109 | 0.108 | 0.109 | 0.108 | 0.107 | 0.107 | 0.104 | 0.104 | 0.103 | 0.102 | 0.105 | 0.103 | 0.104 | 0.104 | 0.103 | 0.101 |
> | RainPro-8 Positive Forecast (%) | 0.130 | 0.146 | 0.143 | 0.133 | 0.145 | 0.195 | 0.182 | 0.198 | 0.182 | 0.204 | 0.199 | 0.185 | 0.218 | 0.206 | 0.191 | 0.227 |
> | RainPro-8   FBI | 1.189 | 1.356 | 1.305 | 1.232 | 1.359 | 1.835 | 1.754 | 1.896 | 1.758 | 1.992 | 1.896 | 1.790 | 2.100 | 1.976 | 1.863 | 2.243 |
> | MetNet-3* Positive  Forecast (%) | 0.117 | 0.151 | 0.167 | 0.148 | 0.198 | 0.203 | 0.218 | 0.231 | 0.227 | 0.214 | 0.213 | 0.211 | 0.184 | 0.188 | 0.223 | 0.218 |
> | MetNet-3*   FBI | 1.067 | 1.399 | 1.525 | 1.372 | 1.853 | 1.902 | 2.103 | 2.214 | 2.201 | 2.090 | 2.026 | 2.044 | 1.780 | 1.796 | 2.171 | 2.157 |
> | RainPro FBI  Improvement (%) | 11% | -3% | -14% | -10% | -27% | -4% | -17% | -14% | -20% | -5% | -6% | -12% | 18% | 10% | -14% | 4% |

---

> > ### Comment · Reviewer_AAd8 · 2025-11-27
> >
> > I thank the authors for their detailed response regarding the issues raised and for providing additional data to further clarify the model. However, regarding these three main points, I retain the following concerns:
> >
> > 1. The methodological contribution is **primarily engineering optimization, lacking innovation in representation learning**. As the authors stated in the rebuttal, the main contribution of RainPro-8 **does not lie in the architecture**, but rather in its engineering implementation, specifically, reducing the parameter count. This lightweight adaptation and efficiency optimization of an existing SOTA model, MetNet-3, is essentially incremental work at the engineering deployment level. While this holds practical value for the community, ICLR focuses on proposing novel methods in **Learning Representations**. The current contributions are better suited for an application-oriented conference or workshop, rather than the main track of ICLR.
> >
> > 2. **Missing baselines**. The authors claim to surpass existing deep learning nowcasting models, but the comparison subjects are biased. In the SEVIR experiments, the authors primarily compare against general video prediction models, STRPM, rather than recognized, expert-validated SOTA generative precipitation forecasting models in the meteorological domain. Using **unavailability of code** as a justification for refusing to compare with these strong baselines—while simultaneously claiming SOTA performance—is not rigorous. By excluding direct comparisons with generative precipitation models, the paper lacks a critical evaluation of forecasting skills against this key category of methods.
> >
> > 3. **Long-term prediction bias increases, while short-term forecasting lacks detail**, rendering its practicality questionable. The data provided in the rebuttal confirms my previous concerns, for prediction horizons exceeding 6 hours, specifically at the 8.0h mark, RainPro-8’s FBI metric (2.243) is higher than the baseline MetNet-3* (2.157). This indicates that its over-forecasting problem in the long term is actually more severe, contradicting the claim that the model is overall more balanced. Furthermore, as a classification model, RainPro-8 inevitably suffers from image blurriness and loss of detail in short-term forecasting (<2h) compared to Generative Models. Although the model achieves balance on certain aggregated numerical metrics, the characteristic of **large long-term bias and blurry short-term details** limits its guidance value in real-world weather forecasting scenarios.
> >
> > In the rebuttal, the authors clarified that the core contribution of RainPro-8 lies in training strategies and inference efficiency, rather than architectural design. While I acknowledge the value of this work as a lightweight, open-source version of MetNet-3 **in terms of engineering reproduction and lowering computational barriers**. After carefully reviewing the paper and rebuttal data, I believe the work still has deficiencies regarding methodological novelty, fairness of baseline comparisons, and reliability of long-term forecasting. Therefore, I will keep my score.

---

> > > ### Author Response · Authors · 2025-11-27
> > >
> > > We thank the reviewer for the continued engagement and for clearly articulating the remaining concerns.
> > >
> > > 1. Contributions
> > > While RainPro-8 leverages the MetNet-3 architecture, we respectfully emphasize that its contributions extend beyond engineering optimization. RainPro-8 introduces a **unified**, **single-pass**, **multi–lead-time probabilistic** forecasting framework that integrates **multiple data sources** and incorporates a **precipitation-specific training objective**. Given that our submission is to the **Applications to Physical Sciences track**, we believe that the combination of **empirical performance**, **accessibility**, and **domain impact** aligns well with the track’s objectives, even if certain representation-learning components are incremental.
> > >
> > > 2. Baselines and generative models
> > > We respectfully disagree that our choice of baselines is biased. If the reviewer can point us toward any open-source generative precipitation models or other models they believe should be included, we would gladly evaluate them. Our **SEVIR baselines follow those used in DiffCast (CVPR 2024)**, which reports state-of-the-art results. Regarding **NowcastNet and DGMR**, we reiterate that these methods **focus solely on short-term, radar-only nowcasting**, whereas **RainPro-8** targets **8-hour, multi-source, probabilistic** precipitation forecasting to **better handle the uncertainty in the later lead-times**. Additionally, because these models are not open-sourced, many essential implementation details cannot be reproduced, making any attempted re-implementation inherently unreliable and therefore not a fair baseline. We note that our SEVIR evaluation already includes comparisons with generative models and the **RainPro-2R** variant, built on the same framework, architecture, and loss, **outperforms both deterministic and generative baselines**. While closed-source generative models cannot be assessed, **we compare against generative methods wherever possible and invite the reviewer to recommend feasible baselines**.
> > >
> > > 3. Performance evaluation
> > > We respectfully disagree with the assertion that the model’s practicality is questionable. RainPro-8 offers a single deep learning model capable of high spatio-temporal resolution precipitation forecasts up to 8 hours ahead, and **our evaluations consistently show improvements across metrics and lead times**. The blurriness observed in short-term forecasts arises from the well-known double-penalty effect rather than from the classification paradigm itself, reflecting a recognized **trade-off between pixelwise accuracy and structural similarity** (Yan et al., 2024). This effect is also present in state-of-the-art models such as MetNet-3.
> > > Regarding **FBI**, we note that it is **not considered a forecast skill metric** in precipitation forecasting. Standard skill metrics **(CSI and FSS) balance true positives, false positives, and false negatives**, and are emphasized across major precipitation-forecasting benchmarks. RainPro-8 consistently achieves higher CSI and FSS than MetNet-3* across nearly all lead times, with **further gains relative to operational and open-source models**. Although RainPro-8 slightly overpredicts at the final timestep compared to MetNet-3*, it delivers higher overall skill, indicating **more accurate detection of precipitation events**. Moreover, as shown in Figure 8, RainPro-8 exhibits **less overprediction than MetNet-3\* for most lead times**.
> > >
> > > We hope these clarifications address the reviewer’s remaining concerns and are taken into consideration for the final decision. Our goal is to provide a high-impact, open, and resource-efficient probabilistic forecasting system that advances accessibility and practical deployment within the weather research community.

---

> ### Comment · Reviewer_iBcq · 2025-11-18
>
> Many thanks to the authors for responding on the comments! While I am reading the rest of the comments, I would like to follow up on the Weakness 3 specifically as I believe it is highly relevant to the overall concern about the experimental performance.
>
> "actually, these qualitative images are not representative, and while in some cases, including some of the ones shown, there is some degree of overprediction, this is not true overall,"
> If the examples picked as case studies are not representative, could the authors point at the scenarios where the claim is clearly met or offer some explanation why this discrepancy could happen? Because while it might be the case that these qualitative results do not correspond to the best outcomes, there should be some explanation why the method performs better in the quantitative evaluation.
>
> Furthermore, I understand the metrics in the table for Weakness 3 are presented as aggregate, as opposed to only for the heavy precipitation, so isn't it the case that Table 1 and Figure 8 are more relevant to the concern that this table?

---

> > ### Author Response · Authors · 2025-11-26
> >
> > Thank you for the engagement during the rebuttal phase.
> >
> > To clarify the confusion: the table does not present aggregate FBI values, it specifically reports values at the 10 mm/h threshold. We included them in tabular form to more clearly show the corresponding values, the FBI improvement of RainPro-8 over MetNet-3*, and how the FBI values were computed for both methods based on total observations and positive predictions at each lead time. The table, consistent with Figure 8, indicates that RainPro-8 generally reduces overforecasting relative to MetNet-3* at most lead times, but not specifically at the final one (8 h), which is the lead time shown in the qualitative visualizations. However, skill and inference speed are still optimal in RainPro-8 compared to MetNet-3*.
> >
> > Regardless, after reviewing the test dataset, we identified more cases where MetNet-3* exhibits a higher FBI (more overprediction) and lower CSI (less skill) than RainPro-8 at the 10 mm/h threshold (heavy precipitation) that we will include in the final version.

---

### Author Response · Authors · 2025-12-03

We thank the reviewers for their detailed feedback. RainPro-8 addresses key challenges in 8-hour high-resolution probabilistic precipitation forecasting by combining multi-source data (radar, satellite, NWP) with a single-pass, multi-lead-time prediction framework and a precipitation-specific ordinal-consistent loss, enabling coherent probabilistic outputs, improved calibration for extreme events, and efficient inference with only 20% of MetNet-3’s parameters and x48 faster runtime.

We here summarise our main responses and additional information:

* Novelty: RainPro-8 introduces novel concepts: namely, single-pass predictions and ordinal-consistent loss, improving accuracy, uncertainty quantification, and operational efficiency - thereby offering a much more lightweight and practically feasible model than existing MetNet-3.
* Baselines: We compare RainPro-8 against both operational and deep-learning models. Generative models are included in the SEVIR experiments that do not provide all data sources via RainPro-2R, where we again observe the benefit of our architecture, single-pass predictions, training strategy, and probabilistic framework in a radar-only 2-hour setup. Models like DGMR and NowcastNet focus on short-term radar-only forecasting, and cannot be directly evaluated in our 8-hour task due to unavailable code and unreproducible training, architecture, and data-processing details.
* Bias: RainPro-8 generally does NOT overpredict, as evidenced also by forecast skill metrics (CSI & FSS) that remain higher than for competitors, considering both false positives and false negatives - still, at some lead times RainPro-8 slightly overpredicts compared to MetNet-3*. We will report additional such information in the final version to substantiate that RainPro-8 offers superior performance while maintaining comparable bias values across lead times.

In conclusion, RainPro-8 provides a robust, resource-efficient, and reproducible deep learning model that advances machine learning from complex multi-source data, with demonstrated benefit for high-resolution probabilistic precipitation forecasting, making it a strong fit for acceptance in the applications to physical sciences topic.

---

### Meta-Review · Area_Chair_ojNs · 2026-01-04

**Summary:**

The paper contributes an open-source and efficient model for high-resolution and probabilistic precipitation forecasting for Europe and at 8-hours lead time.
The reviewers underscored the importance of bridging the gap between nowcasting and medium-range forecasting, praised the extent and rigor of the experiments and ablation studies, but highlighted the limited technical novelty of the work. Another limitation is the model's limited applicability to any region, due to the need of data sources not available everywhere.
The decision of accepting the paper is based on its potential impact. Indeed, the introduction of an ordinal loss combined with simple modifications to the MetNet-3 architecture and the use of multiple data sources unlocks x48 faster inference and x1900 cheaper training compared to MetNet-3, at a comparable accuracy.

**Reviewer Concerns:**

Apart from the limited technical novelty and applicability outside Europe, the reviewers raised the following major concerns that were sufficiently addressed during the rebuttal:
1. **Lack of comparison with GAN-based models.** The authors highlighted the difficulty and unsuitability of comparing with NowcastNet and DGMR, because, unlike RainPro-8, they are closed-source, expensive, radar-only and nowcast models. The authors also pointed to the evaluation in Section 4.5 where performance of standard generative baselines is evaluated.
2. **Contradictory results (between aggregated metrics and prediction plots) and lack of extreme-specific results.** The contradiction is due to the selected predictions not being representative of the overall performance of the model. Further, the paper reports several established metrics, including the CSI metric for assessing the ability of forecasting extremes.
3. **RainPro-8's large long-term bias and blurry short-term details.** The quantitative evaluation indicates that RainPro-8 achieves state-of-the-art performance across almost all lead times. The principal baseline, MetNet3, also shows large long-term bias and the visual blurriness is a common issue of optimizing pixel-based metrics.

**Reviewer Scores:**

Based on the visible discussion, none of the reviewers would have luckily changed their scores.

---

### Decision · Program_Chairs · 2026-01-26

Accept (Poster)